# Quality of Assessment Tools for Aphasia: A Systematic Review

**DOI:** 10.3390/brainsci15030271

**Published:** 2025-03-03

**Authors:** Francescaroberta Panuccio, Giulia Rossi, Anita Di Nuzzo, Ilaria Ruotolo, Giada Cianfriglia, Rachele Simeon, Giovanni Sellitto, Anna Berardi, Giovanni Galeoto

**Affiliations:** 1Department of Human Neurosciences, Sapienza University of Rome, Viale dell’Università, 30, 00185 Rome, Italy; francescaroberta.panuccio@uniroma1.it (F.P.); giu.rossi@uniroma1.it (G.R.); anitadinuzzoe@gmail.com (A.D.N.); ilaria.ruotolo@uniroma1.it (I.R.); giada.cianfriglia@gmail.com (G.C.); giovanni.sellitto@uniroma1.it (G.S.); anna.berardi@uniroma1.it (A.B.); 2Department of Public Sciences and Infectious Diseases, Sapienza University of Rome, 00185 Rome, Italy; 3Department of Neuroscience, Rehabilitation, Ophthalmology, Genetics and Maternal Child Health (DINOGMI), University of Genoa, 16126 Genoa, Italy; rachele.simeon@edu.unige.it; 4IRCCS Neuromed, Via Atinense, 18, 86077 Pozzilli, Italy

**Keywords:** aphasia, psychometric properties, questionnaire, reliability, systematic review, tools, validation

## Abstract

**Background/Objectives**: Aphasia is a neurological condition affecting the ability to understand and/or express language fluently and accurately, and can occur following stroke, traumatic injuries, or other brain pathologies. The aim of the following study was to provide clinicians and researchers information regarding the existing assessment tools to assess aphasia. **Methods**: For this Systematic Review, PubMed, CINAHL, Web of Science, and Scopus were searched for articles published up to August 2024. Authors independently identified eligible studies based on predefined inclusion criteria and extracted data. The study quality and risk of bias were assessed using the Consensus-based Standards for the Selection of Health Measurement Instruments (COSMIN) checklist. **Results:** Of the 1278 publications identified and screened, 238 studies fell within the inclusion criteria and were critically reviewed, and 164 assessment tools were found and divided into 8 main domains; the most used tools were the Language Screening Test (LAST), the Stroke and Aphasia Quality of Life Scale-39 (SAQOL-39), the Oxford Cognitive Screen (OCS), and the Token test. **Conclusions**: This review has emphasized the need for agreement among researchers as to which tool must be studied or adapted to other national contexts to develop universal norms and standards.

## 1. Introduction

Aphasia is defined as the “partial or complete loss of linguistic abilities, i.e., linguistic comprehension or expression, or both, resulting from damage to the language areas of the brain and not attributable to speech difficulties, i.e., disorders of the mechanical processes of language” [1]. It is considered as an acquired language disorder, as it can be caused by different brain damages and disorders, such as cerebrovascular accidents (CVAs); the progressive deterioration of brain tissue such as Alzheimer’s and vascular dementia; or from direct injuries, such as Traumatic Brain Injuries (TBIs) and brain tumors [2,3]. The symptoms can range from mild impairment to the complete loss of any fundamental components of language (semantic, grammar, phonology, morphology, and syntax) [4]; in the forms resulting from a focal lesion, patients typically exhibit focal neurological deficits such as hemiplegia, hemianesthesia, homonymous hemianopsia, etc., while degenerative forms of aphasia are often associated with dementia [1].

Aphasic syndromes depend on the location of the brain damage: the fluent aphasia (or receptive aphasia) syndromes are primarily associated with damage to the posterior areas of the left hemisphere (e.g., the Wernicke area, angular gyrus, etc.) and are characterized by fluent speech with little meaningful content, as well as by the inability to understand written and/or spoken words. In contrast, the non-fluent aphasia (or expressive aphasia) syndromes are associated with damage to the anterior part of the left hemisphere of the brain (e.g., Broca area, premotor and motor cortex, etc.) and are characterized by limited and effortful spontaneous speech, and grammatically simple and inaccurate sentences [1].

Speech and language are fundamental functions both in social relationships and in intellectual activities (e.g., study, work, etc.). In fact, aphasia generally causes significant changes in a patient’s quality of life (QoL), leading to impairments in language skills (e.g., verbal expression, auditory comprehension, reading, and writing) which also negatively affect self-image and family, social, and work roles [5].

According to a 2022 scientific paper [6], about 15 million people worldwide are affected by aphasia, particularly older adults who are at a higher risk for stroke. However, its reported incidence and prevalence vary widely, likely due to its variety of conditions, the heterogeneity of the population affected, and the different methodologies and criteria used to determine and assess the presence of aphasia [7]. Until recently, as a result of the model of organization of oral language proposed by Wernicke and Lichtheim, the classification of language deficits was mainly based on the dichotomy between receptive and expressive aphasia, and/or anterior and posterior lesions [8]. However, the great majority of aphasic patients do not present exclusively motor or sensory deficits.

In clinical practice, it is essential that we have objective measurement tools that allow healthcare professionals to assess the extent and daily impact of aphasia symptoms in the patient, considering the linguistic, cognitive, relational, and emotional aspects of the disorder. Upon reviewing the scientific literature, it becomes evident that there are numerous assessment tools for aphasia which often lead to inconsistencies and make it difficult to compare study results. The importance of standardized approaches to outcome measurement in research trials has increasingly been recognized. In the field of stroke rehabilitation, initiatives such as the Stroke Recovery and Rehabilitation Roundtable (SRRR) have provided consensus-based recommendations for the measurement of sensorimotor recovery and stroke outcomes [9]. The same efforts have been made to establish standardized recommendations for measuring outcomes in aphasia. In 2022, the Research Outcome Measurement in Aphasia (ROMA) developed a Core Outcome Set (COS) through a series of international consensus studies involving individuals with aphasia, their families, clinicians, and researchers [10]. This initiative identified five core outcome constructs for aphasia research: (1) communication, (2) language, (3) quality of life, (4) emotional well-being, and (5) patient-reported satisfaction with or impact of treatment. Each of these constructs represents a critical dimension of aphasia rehabilitation and its impact on individuals’ lives. The recommended outcome measurement instruments (OMIs) include the Western Aphasia Battery—Revised (WAB-R) for language, the General Health Questionnaire—12 (GHQ-12) for emotional well-being, and the Stroke and Aphasia Quality of Life Scale (SAQOL-39) for quality of life.

ROMA-COS highlights the need for consensus on the tools used to measure these outcomes, as well as the importance of selecting instruments with robust psychometric properties. The COnsensus-based Standards for the selection of health Measurement Instruments (COSMIN) checklist provides a standardized approach to assessing the methodological quality of instruments, ensuring that evaluations are scientifically rigorous and based on transparent criteria. This highlights the need for standardized tools that comprehensively evaluate the different dimensions of aphasia. The aim of this review was to identify currently available measurement tools (such as scales, tests, and questionnaires) that assess various aspects of communication and cognition in individuals with aphasia, and in which psychometric properties are documented in scientific papers and manuals.

## 2. Materials and Methods

This study was conducted by the R.E.S. (Ricerca Evidenza e Sviluppo) research group from the “Sapienza” University of Rome; in the last few years, R.E.S. was involved in carrying out systematic reviews and validation of assessment tools studies [11,12,13,14,15,16,17,18,19].

### 2.1. Protocol and Registration

After registering the protocol on the Prospero website, the international prospective register of systematic reviews, available at the registration number CRD42024589980, this review was carried out in accordance with the 27-item Preferred Reporting Items for Systematic Reviews and Meta-Analyses (PRISMA) checklist [20,21,22,23] and COnsensus-based Standards for the selection of health Measurement Instruments (COSMIN) methodology for systematic reviews of Patient-Reported Outcome Measures (PROMs) [20].

### 2.2. Information Sources

This systematic review focused on the studies that evaluated the psychometric properties of assessment tools exploring aphasia; excluded tests were not necessarily deemed invalid, but because they did not meet the predefined criteria of providing explicit information regarding the analysis and reporting of psychometric properties.

The following electronic databases were systematically searched until August 2024: MEDLINE (via PubMed), CINAHL, SCOPUS, and Web Of Science. The reviewers chose to use the databases mentioned above as they index only journals that follow the peer review process, to keep the methodological quality of the study high; this is why we also chose to not use the gray literature. No restrictions were applied to the publication period, the country in which the study was conducted, or the age of patients.

### 2.3. Inclusion and Exclusion Criteria

To be included in the following review, manuscripts had to address the following:(1)Validation studies and cross-cultural adaptation studies;(2)Studies about evaluating aphasia;(3)Studies about tests, questionnaires, and self-reported and performance-based outcome measures.

Both diagnostic tools and outcome measures were included in this review. Diagnostic tools are designed to identify and classify patterns of aphasia based on linguistic functions and are typically used to diagnose and/or characterize the nature of aphasia. In contrast, outcome measures are employed to detect changes over time, particularly those resulting from therapeutic interventions, thereby assessing the effectiveness of treatments.

Trials or studies that evaluated the effectiveness of interventions where a questionnaire is used as an endpoint (without studying the measurement properties) were excluded.

### 2.4. Search Methods for Identification of the Studies and Electronic Searches

Studies were identified for inclusion through individualized systematic searches of four electronic databases. All potential studies were identified by four reviewers.

The search strategy was developed by the review’s primary reviewer and, following consultation with an expert, using guidance from relevant past reviews. A combination of terms and keywords was used for each database:(1)Medline (via PubMed): (assessments or evaluation or screening or test or measurement) AND (aphasia)) AND (reliability and validity);(2)CINAHL (via EBSCO): (reliability and validity) AND (assessment or evaluation or screening or test or measurement) AND aphasia;(3)Web of Science (via EBSCO): (reliability and validity) AND (assessment or evaluation or screening or test or measurement) AND aphasia;(4)Scopus: TITLE-ABS-KEY (reliability AND validity) AND TITLE-ABS-KEY (assessments OR evaluation OR screening OR test OR measurement) AND TITLE-ABS-KEY (aphasia).

### 2.5. Study Selection

From the results of the database research titles, keywords and abstracts were independently screened by one speech therapist and one occupational therapist (98% agreement). After the first screening, the primary reviewer selected the relevant studies and assessed them against the inclusion criteria; then, a second reviewer cross-checked so that the studies that did not fit the inclusion criteria were systematically excluded and others that appeared pertinent were identified. The studies that met the criteria were then subject to a full text review, and an initial stratification of the assessment tools was set up in the following evaluation areas: language-specific assessment tools, Quality of Life (QoL) assessment tools, cognitive assessment tools, acute stroke assessment, tools to be administered to caregivers and healthcare professionals who are in charge of people with aphasia (PWA), anosognosia assessment tool, auditory and perceptual assessment tool, and praxis assessment tool.

### 2.6. Data Collection, Data Items, and Assessment of Risk of Bias

Two reviewers independently extracted the results of the included studies; the following data were extracted from each article: author name (year of publication), language, population, number of sample size, mean age ± standard deviation, and assessment categories.

Assessment tools reported within each publication were recorded and categorized for comparison. Study quality and risk of bias were assessed using the COSMIN checklist, a recommended resource for planning studies that aim to evaluate the measurement properties of existing patient-recommended outcome measures (PROMs) [20]. To ensure consistency and reliability, two researchers, not involved in the study selection process, independently assessed adherence to the COSMIN checklist. Any discrepancies were resolved through a meeting with the group coordinator.

The COSMIN Risk of Bias tool consists of two parts. Part A evaluates the reliability or measurement error of the outcome measure instrument, while part B examines the trustworthiness of the study results by assessing the risk of bias. Each criterion on the checklist is rated on a 4-point scale: ‘very good’, ‘adequate’, ‘doubtful’, and ‘inadequate’. The COSMIN study design checklist includes 10 sections. The first section, which covers general recommendations for designing a study on measurement properties, is applicable to all studies and includes essential standards to be considered in any study design. The remaining sections provide specific standards for studies, focusing on each of the nine measurement properties, specifically:-Content validity, the degree to which the content of an instrument is an adequate reflection of the construct to be measured, with the aim to investigate the relevance and comprehensiveness of multi-item measures. It can be qualitatively evaluated by an independent panel of experts to avoid risk of bias. An additional aspect of the content validity is represented by face validity, defined as the degree to which a measurement instrument looks as an adequate reflection of the instrument to be measured.-Construct validity, defined as the degree to which the scores of an instrument are an adequate reflection of the dimensionality of the construct to be measured, regarding internal relationships, relationships with scores of other instruments, or differences between relevant groups. It can be divided into three subtypes: (1) structural validity, (2) hypothesis testing, and (3) cross-cultural validity.-Internal consistency, which represents the level to which items belonging to an assessment tool assess the same construct. It concerns the so-called interrelatedness among the items.-Cross-cultural validity/measurement invariance, defined as the degree to which the performance of the items on a translated or culturally adapted tool is an adequate reflection of the performance of items in the original version. It is very important after the translation of a questionnaire.-Reliability, the degree to which the measurement is free from measurement error. It varies depending on issues that include the instrument under investigation, the evaluators, and the patients under study. It can be divided into four subtypes: (1) test–retest reliability, when measurements are repeated over time; (2) inter-rater reliability, when they are conducted by different evaluators but on the same occasion; (3) intra-rater reliability, when they are conducted by the same evaluator but on different occasions; and (4) internal consistency, when different sets of items from the same tool are employed.-Criterion validity, defined as the degree to which the scores of a measurement instrument are an adequate reflection of a gold standard. It can be divided into two main sides: (1) concurrent validity and (2) predictive validity.-Measurement error, concerning the systematic and random error of a patient’s score that is not attributed to true changes in the construct to be measured, corresponding to the difference between an amount that can be measured and its true value.-Hypothesis testing, used to determine whether an instrument accurately measures a construct by comparing its scores to hypotheses based on theoretical expectations;-Responsiveness, the ability of an instrument to detect change over time in the construct to be measured, reflecting if the clinical status of patients has changed over time. When a tool is shown to be responsive to change if patients change on the construct of interest, their scores on the measurement tool assess this construct change accordingly.

## 3. Results

### 3.1. Study Selection

The search ended in August 2024, identifying 1278 total results from the research strategy. After removing duplicates, 552 articles were screened by reading the titles and abstracts, resulting in 726 papers that were screened further by reading the full text. The articles reviewed were published between 1963 and 2024. From the 447 excluded studies, 321 met the inclusion criteria, which led to the extraction of 164 assessment tools. The selection process is reported in the flowchart (Figure 1), according to the PRISMA guidelines for reporting systematic reviews and meta-analyses [20,21].

The obtained assessment tools can be divided into specific evaluation areas: language-specific (n = 80), Quality of Life (QoL) (n = 29), cognitive (n = 23), acute stroke assessment (n = 1), caregivers and healthcare professionals assessment (n = 4), anosognosia (n = 1), auditory and perceptual (n = 1), and praxis (n = 1). Sixteen assessment tools are, instead, multidimensional, since they investigate one or more domains among those listed.

The tools identified in this review were categorized into three main groups: diagnostic tools (n = 78), screening tools (n = 16), and outcome measures (n = 65). Additionally, seven tools were classified as unspecified due to insufficient information regarding their primary purpose or application. Table 1 presents a detailed breakdown of the articles corresponding to each category, providing an overview of the tools and their intended use within clinical and research contexts.

### 3.2. Study Characteristics

For each of the included studies, the following characteristics were collected and included in Table 2: author name (year of publication), language, population, number of sample size, mean age ± standard deviation, and assessment categories.

Of the included studies, the sample size varied from a minimum of 10 [24,25,26] to a maximum of 955 [27]. The mean age varies, ranging from 19 [28] to 87.8 ± 6.5 [29]. The most represented languages were English (n = 75 assessment tools), French (n = 30 assessment tools), German (n = 15 assessment tools), Portuguese (n = 12 assessment tools), and Dutch (n = 10 assessment tools).

**Table 2 brainsci-15-00271-t002:** Descriptive information of included studies.

Assessment Tool	Author (Year)	Language	Population	Sample Size (n)	Mean ± SD Age	Cronbach’s α	ICC	Construct/Concurrent	Assessment Category
Aachen Aphasia Test (AAT)	Huber W. (1984) [30]	German	Stroke	n.a.	n.a.	n.a.	n.a.	n.a.	Language
	Pracharitpukdee N. (2000) [31]	Thai	Healthy population, Stroke, Other	HP: 120 S: 125 O: 60	n.a.	n.a.	n.a.	n.a.	
	Miller N. (2000) [32]	English, German	Healthy population, Stroke	228	HP: 51.8 S: 62.7	0.72–0.99	n.a.	n.a.	
	Lauterbach M. (2008) [33]	Portuguese	Stroke, TBI, brain tumors or arteriovenous malformations	278	57.7	0.90	n.a.	n.a.	
	Luzzatti C. (2023) [34]	Italian, Dutch, German	Stroke	674	59.8	0.75–0.99	n.a.	TT	
Aachener Aphasie Bedside Test (AABT)	Biniek R. (1992) [35]	German	Stroke	82	n.a.	n.a.	n.a.	n.a.	Language
	Muò (2021) [36]	Italian	Stroke, TBI, brain tumors	248	70.6	n.a.	0.69–0.99	AAT	
Aachener Sprachanalyse (ASPA)	Barthel G. (2006) [37]	German	Stroke	n.a.	n.a.	n.a.	n.a.	n.a.	Language
Abbey Pain Scale (APS)	Abbey J. (2004) [38]	English	Dementia	n.a.	n.a.	0.74	n.a.	n.a.	Language
	Storti M. (2009) [39]	Italian	Dementia	n.a.	n.a.	n.a.	n.a.	n.a.	
	Takai Y. (2010) [40]	Japanese	Dementia, Alzheimer	171	85.4 (8)	0.64–0.72	0.66–0.85	VDS	
	Neville C. (2013) [41]	English	Dementia	126	85.2 (6.6)	0.71	n.a.	CNPI	
	Gregersen M. (2016) [42]	Danish	Stroke	50	70	0.52	0.84	VRS	
Acute Aphasia Screening Protocol (AASP)	Crary M.A. (1989) [43]	English	Stroke	48	69.42	n.a.	n.a.	n.a.	Language, Cognitive
Addenbrooke’s Cognitive Examination (ACE)	Hodges J.R. (2017) [44]	English	Corticobasal Syndrome, Dementia	CBS: 21 D: 23	68.8	n.a.	n.a.	n.a.	Cognitive
	Gaber T.A. (2011) [45]	English	Stroke	73	72	n.a.	n.a.	n.a.	
	Elamin M. (2016) [46]	English	Stroke	71	62.2 (5.4)	n.a.	n.a.	n.a.	
Mini—Addenbrooke’s Cognitive Examination (Mini-ACE)	Hsieh S. (2015) [47]	English	Healthy population, FTD, Dementia, Alzheimer, Corticobasal Syndrome	164	HP: 67.4 (6.4) FTD: 61.3 (10.7) A; 64 (8.3) D: 66.4 (8.7) CBS: 65.3 (7.6)	0.83	n.a.	MMSE FTDFRS	Cognitive
American Speech-Language and Hearing Association (ASHA-FACS)	de Carvalho I.A.M. (2008) [48]	Portuguese	Healthy Population, Alzheimer	n.a.	n.a.	0.95	0.99	ADAS	Language
	Muò R. (2015) [49]	Italian	Stroke, TBI	180	n.a.	0.84	n.a.	FIM	
Amsterdam-Nijmegen Test for Everyday Language (ANELT)	Blomert L. (1994) [50]	Dutch	Stroke	260	65	0.90	n.a.	n.a.	Language
	Ruiter M.B. (2011) [51]	Dutch	Healthy population, Stroke	HP: 20 S: 10	58	n.a.	0.66–0.93	n.a.	
	Ruiter M.B. (2022) [52]	Dutch	Healthy population, Stroke	HP: 31 S: 17	HP: 49 (17) S: 56 (13)	0.87	0.42–0.80	n.a.	
	Wong W.W.S. (2024) [53]	Chinese	Stroke, Other Neurological Conditions	S: 56 O: 100	n.a.	0.84–0.89	n.a.	FIM AAT LCF	
An Object and Action Naming Battery (An O&A Battery)	Edmonds (2012) [54]	English	Stroke	91	22.5	n.a.	n.a.	An O&A	Language
	Spezzano L.C. (2013) [55]	Portuguese	Healthy Population	G1: 50 G2: 50	G1: 56.7 (17.9) G2: 53.3 (15.2)	n.a.	n.a.	n.a.	
Aphasia and stroke therapeutic alliance measure (A-STAM)	Lawton M. (2019) [56]	English	Stroke	34	63.2	0.92	0.93–0.97	WAI ADRS	Caregivers and Healthcare professionals
AphasiaBank Stimuli	Boyle M. (2015) [57]	English	Stroke, TBI	10	n.a.	n.a.	−0.08–0.96	n.a.	Language
Aphasia Check List (ACL)	Kalbe E. (2005) [58]	German	Stroke, tumor, TBI, encephalitis	154	58.0 (17.1)	0.40–0.88	0.55–0.91	AAT	Language and Cognitive
	Zadeh A.M. (2021) [59]	Iranian	Stroke	20	50	0.77–0.80	0.98	n.a.	
Aphasia Communication Outcome Measure (ACOM)	Hula W.D. (2015) [60]	English	Stroke, Tumor, Radiation necrosis	329	60.5	n.a.	n.a.	ASHA-FACS PICA BDAE	Quality of Life
Aphasia Impact Questionnaire 21 (AIQ)	Swinburn K. (2019) [61]	English	Stroke	137	65.86 (14.60)	0.77–0.92	n.a.	BOSS	Quality of life
	Yaşar E (2022) [62]	Turkish	Stroke	30	54.32 (9.08)	0.91	n.a.	SAQOL-39	
Aphasia Rapid Test (ART)	Azuar C. (2013) [63]	French	Stroke	91	63.96 (19.3)	n.a.	n.a.	n.a.	Acute Stroke Assessment
	Panebianco M. (2019) [64]	Italian	Stroke	143	73.73	n.a.	n.a.	n.a.	
	Buivolova O. (2021) [65]	Russian	Stroke	105	58.9	0.76	n.a.	Vasserman’s scale TT	
	Kavakci M. (2022) [66]	Turkish	Stroke	30	64.43	n.a.	n.a.	n.a.	
Aphasic Depression Rating Scale (ADRS)	Benaim C. (2004) [67]	French	Stroke	52	60 (13)	n.a.	n.a.	HDRS	Quality of Life
Apraxia of Speech Rating Scale (ASRS)	Strand E.A. (2014) [68]	English	Stroke	134	67.90 (2.21)				Language
	Wambaugh J.L. (2019) [69]	English	Stroke, TBI	28	n.a.	n.a.	0.03–0.95	ASSIDS	
	Hybbinette H. (2021) [70]	Swedish	Stroke	10	46.1 (11.84)	n.a.	0.42–0.69	n.a.	
	Duffy J.R. (2023) [71]	English	Healthy Population, Stroke	308	70	n.a.	0.98	n.a.	
	Santos D.H.N.D. (2023) [72]	Portuguese	Healthcare professionals	13	n.a.	n.a.	n.a.	n.a.	
Assessment of Living with Aphasia (ALA)	Simmons-Mackie N. (2014) [73]	English	Stroke, TBI, Other	101	64.54	0.59–0.89	0.86	BOSS CAPD VASES SAQOL-39	Quality of Life
	Guo Y.E. (2017) [74]	English	Stroke	66	61.4	n.a.	0.99	n.a.	
Assessment of Communicative Effectiveness in Severe Aphasia (ACESA)	Cunningham R. (1995) [75]	English	Stroke	20	62.9	n.a.	0.86–1.00	n.a.	Quality of Live
Auditory Comprehension Test for Sentences (ACTS)	Klor B.M. (1980) [76]	English	Healthy population	180	n.a.	n.a.	n.a.	n.a.	Language
	Flanagan J.L. (1997) [77]	English	Healthy population	31	63.74 (7.4)	n.a.	n.a.	n.a.	
Auditory-Perceptual Rating of Connected Speech in Aphasia (APROCSA)	Casilio M. (2019) [78]	English	Stroke	12	25.5 (3.3)	n.a.	0.08–0.95	WAB-R	Auditory-perceptive
Augmentative and Alternative Communication Assessment Questionnaire (AAC)	Petrosyan T.R. (2022) [79]	Armenian	Neurological disorders	210	25 (6.6)	0.86–0.96	n.a.	FCP	Language, Cognitive, Quality of Life
Azeri aphasia screening test	Salehi S. (2016) [80]	Iranian	Stroke, TBI	32	64	0.91	0.88–0.97	n.a.	Language
Batterie d’évaluation de la compréhension syntaxique (BEPS)	Bourgeois M.E. (2019) [81]	French	Stroke	130	65.3	n.a.	0.96	n.a.	Language
	Coulombe V. (2021) [82]	French	Stroke, Healthy Population	PS: 6 PPA: 6 PPAOS: 3 HP: 14	PS: 64.5 (11.3) PPA: 77.3 (7.61) PPAOS: 72.7 (2.08) HP: 51.5 (15.53)	0.51–0.92	0.98	DVL 38 BECLA	
Bedside Aphasia Battery (BAB)	Sivagnanapandian D. (2022) [83]	Indian	Stroke	105	60.9	0.99	n.a.	n.a.	Language
Bedside Aphasia Screening Test (BAST)	Cruz A.L. (2014) [84]	Portuguese	Stroke	112	67.49 (11.83)	0.91–0.98	n.a.	AQ	Language, Cognitive
Behavioural Outcomes of Anxiety questionnaire (BOA)	Eccles A. (2017) [85]	English	Stroke	111	69.7	0.39–0.66	n.a.	HADS GAD-7	Quality of Life
Bilingual Aphasia Test (BAT)	Amberber A.M. (2011) [86]	Rarotongan	Stroke	n.a.	n.a.	n.a.	n.a.	n.a.	Language
	Gomez-Ruiz I (2011) [87]	Spanish	Alzheimer	45	74.60	n.a.	n.a.	n.a.	
	Peristeri E. (2011) [88]	Greek	Stroke	9	65.6 (16.7)	n.a.	n.a.	BDAE-SF	
	Amberber A.M. (2012) [89]	English	Dementia	n.a.	n.a.	n.a.	n.a.	n.a	
	Krishnan G. (2017) [90]	Malayalam (India)	Stroke	22	47.13	n.a.	0.99–1.00	n.a.	
Birmingham Cognitive Screen (BCoS)	Pan X. (2015) [91]	Chinese	Atherosclerosis, Cardioembolism, Artery Occlusion	231	65.70 (8.99)	n.a.	n.a.	0.92–0.99	Language and Cognitive
	Kong A.P.H. (2018) [92]	Chinese	Stroke	22	n.a.	n.a.	n.a.	n.a.	
	Kuzmina E. (2018) [93]	Russian	Stroke	113	70.50	0.71–0.86	n.a.	n.a.	
Basic Outcome Measure Protocol for Aphasia (BOMPA)	Kagan A. (2020) [94]	English	Healthcare Professionals	20	n.a.	n.a.	0.65–0.90	MPC ASR	Language and Quality of Life
Boston Diagnostic Aphasia Examination (BDAE)	Pineda D.A. (2000) [95]	Spanish	Healthy population	156	n.a.	n.a.	n.a.	n.a.	Language
	Fong M.W.E. (2019) [96]	English	Stroke, TBI	355	56.98	n.a.	n.a.	n.a.	
Boston Diagnostic Aphasia Examination—Short Form (BDAE-SF)	Flanagan J.L. (1997) [77]	English	Healthy population	31	63.74 (7.4)	n.a.	n.a.	n.a.	
	Tsapkini K. (2009) [97]	Greek	Healthy Population, Stroke	HP: 100 S: 16	HP: 51.4 (16.6) S: 65.8 (12.5)	n.a.	n.a.	n.a.	
	Del Toro C.M. (2011)	English	Stroke	100	62.9 (12.5)	n.a.	n.a.	n.a.	
Boston Naming Test (BNT)	Del Toro C.M. (2011) [96]	English	Stroke	100	62.9 (12.5)	n.a.	0.77	n.a.	Language
	Aniwattanapong D. (2019) [97]	Thai	Healthy Population, Alzheimer, Mild Cognitive Impairment	HP: 60 A: 60 MCI: 60	HP: 68 (5.7) A: 78.8 (7.1) MCI: 74.8 (6.3)	0.93–0.92	n.a.	n.a.	
	Sachs A. (2020) [98]	English	Acute neurological damage	42	60.3	n.a.	n.a.	n.a.	
Brief Aphasia Evaluation (BAE)	Vigliecca N.S. (2011) [99]	Spanish	Malformations, Stroke, Brain Tumors, Brain Cysts, TBI	109	50.57	0.99	n.a.	n.a.	Language
	Vigliecca. N.S. (2019) [100]	Spanish	Stroke	67	47.24	n.a.	n.a.	n.a.	
Brief Evaluation of Receptive Aphasia (BERA)	Aubinet C. (2021) [101]	French	TBI	62	n.a.	n.a	n.a.	LAST	Language
Brief test of Cognitive-Communication Disorders (BCCD)	Lee M.S. (2020) [27]	Korean	Healthy population, Dementia, Mild Cognitive Impairment, TBI	955	69.50	0.837	n.a.	MMSE	Cognitive
Categorical Naming Test (CNT)	Hwang Y.M. (2021) [102]	Korean	Stroke, Encephalitis, TBI, Epilepsy, Other	333	59.78 (13.91)	0.69–0.85	n.a.	BNT	Language
City Gesture Checklist (CGC)	Caute A. (2021) [103]	English	Healthy population	18	n.a.	n.a.	0.68	n.a.	Praxis
Closed Answers, Pro-speak question, Simple orders, Common object denomination, Audio repetition, Reading, Evoke (CA-PS CARE)	Ferri L. (2021) [104]	Italian	Seizures	20	37.7	n.a.	n.a.	n.a.	Language
Cognitive assessment scale for stroke patients (CASP)	Park K.H. (2017) [105]	Korean	Stroke	33	67.67 (12.95)	0.90	0.99	MMSE	Cognitive
	Benaim C. (2022) [106]	Swedish and French	Stroke	201	63	0.78	0.37–0.89	n.a.	
Core Assessment of Language Processing (CALAP)	Jacquemot C. (2019) [107]	French	Healthy population, Neurological disorders	189	49.1	0.88	0.99	BDAE	Language
Core Lexicon and Microlinguistic Measures	Kim H. (2019) [108]	English	Stroke	11	61.7 (14.7)	n.a.	0.94–0.99	n.a.	Language
Communication Confidence Rating Scale for Aphasia (CCRSA)	Cherney L.R. (2011) [109]	English	Stroke	21	66.7	0.88	n.a.	n.a.	Quality of Life
Communication Outcome after Stroke (COAST)	Long A.F. (2009) [110]	English	Stroke	102	68	0.95	0.90	FAST	Quality of Life and Caregivers
	Bambini V. (2017) [111]	Italian	Cerebrovascular accidents, TBI, Brain Tumors	58	56.15	0.68–0.94	0.69–0.94	AAT	
Carer Communication Outcome after Stroke (Carer COAST)	Long A.F. (2009) [110]	English	Caregivers, Stroke	116	68	n.a.	0.91	COPE	
Communicative Activities Checklist (COMACT)	Aujla S. (2015) [112]	English	Healthy population, Stroke	HP: 75 S: 31	HP: 74 S: 71	0.21–0.83	n.a.	WAB BNT CADL	Language
Communicative Access Measures for Stroke (CAMS)	Kagan A. (2017) [113]	English	Stroke	63	n.a.	n.a.	0.05–0.95	n.a.	Language
Communicative Activity Log (CAL)	Kim D.Y. (2019) [114]	Korean and German	Stroke	50	57.96 (12.82)	0.99	n.a.	n.a.	Language
	Habili M. (2022) [115]	Persian	Stroke	25	59.16	0.92	n.a.	n.a.	
Communicative Competence Scale (CCS)	Brock K.L. (2019) [116]	English	Healthcare professionals	55	52	0.70–0.86	n.a.	n.a.	Quality of Life
Communicative Effectiveness Index (CETI)	Lomas J. (1989) [117]	Swedish	Caregivers, Stroke	n.a.	n.a.				Quality of Life
	Pedersen P.M. (2001) [118]	Danish	Stroke	68	72.6	0.96	n.a.	WAB PICA	
	Moretta P. (2021) [119]	Italian	Stroke	68	n.a.	0.90	0.94	WAB	
	Charalambous M. (2024) [120]	Greek	Caregivers, Stroke	60	S: 67.67 (10.71) S: 47.4 (16.33)	0.95	0.93	ASRS HADS AIQ-21	
Communicative Participation Item Bank (CPIB)	Baylor C. (2017) [121]	English	Stroke	110	60.2 (13.3)	n.a.	n.a.	n.a.	Quality of Life
Community Integration Questionnaire (CIQ)	Dalemans R.J. (2010) [122]	Dutch and English	TBI	150	64.2 (11.1)	0.75	0.96	BI COOP-WONCA	Quality of Life
Comprehensive Aphasia Test (CAT)	Abou El-Ella M.Y. (2013) [123]	Arabic	Stroke, TBI	100	50.5	n.a.	n.a.	n.a.	Language and Cognitive
	Maviş İ. (2022) [124]	Turkish	Stroke, TBI	290	61.07	0.88–0.89	n.a.	ADD	
	Zakariás L. (2022) [125]	Hungarian	Stroke	134	57.6	0.63–0.96	0.28–0.98	WAB TROG-H	
	Kong A.P.H. (2022) [126]	Chinese	Stroke	32	n.a.	n.a.	n.a.	n.a.	
	Jensen B.U. (2024) [127]	Norwegian	Stroke, Healthy population	169	61.8	0.87–0.92	n.a.	n.a.	
Comprehensive Assessment of Reading in Aphasia (CARA)	Thumbeck S.M. (2023) [128]	German and English	Stroke	22	58.6	0.88	n.a.	n.a.	Language
Computerized Language Analysis (CLAN)	Hsu C.J. (2018) [129]	English	Healthy population, Stroke	18	HP: 58.2 S: 57.4	n.a.	n.a.	n.a.	Language
Confrontation Naming Test (CNT)	Vigliecca N.S. (2019) [130]	Spanish	Brain Tumors, Stroke, TBI, Brain Cysts, Other	292	45.41 (20.66)	0.77–0.79	n.a.	n.a.	Memory
Controlled Oral Word Association Test (COWAT)	Ross T.P. (2003) [131]	English	Healthy population	125	20.1 (1.7)	n.a.	0.94–0.99	NAART	Language
Conversation and Communication Questionnaire for People with Aphasia (CCQA)	Horton S. (2020) [132]	English	Stroke	35	n.a.	0.70–0.91	n.a.	n.a.	Quality of Life
Cracow Neurolinguistic Battery of Aphasia Examination (CN-BAE)	Pachalska M. (1995) [133]	Polish	Healthy population, Mental Disorders, Stroke	340	n.a.	n.a.	n.a.	NAI	Language and Cognitive
Cuestionario para la Evaluación Enfermera de las Capacidades Comunicativas en la Afasia (CEECA)	Martín-Dorta W.J. (2023) [134]	Spanish	Healthy population	50	n.a.	0.98	n.a.	BNT NANDA-I NOC	Healthcare professionals
Decision-Making Capacity Assessments (DMCA)	Carr F.M. (2023) [135]	English	Stroke	27	n.a.	0.76	n.a.	n.a.	Quality of Life
Detection Test for Language impairments in Adults and the Aged (DTLA)	Macoir J. (2017) [136]	French	Healthy population, Alzheimer, Stroke, Dementia	602	HP: 63.96 (9.21) A: 77.75 (7.85) S: 68.06 (10.86) D: 76 (7.07)	0.84	n.a.	BNT MT-86 MEC WMS-IV PPT	Language
	Karalı F.S. (2024) [137]	Turkish	Healthy population, Alzheimer	175	62.94 (8.46)	0.84	n.a.	n.a.	
Diagnostic Aphasia Battery (DAB)	Al-Thalaya Z. (2018) [138]	Arabic	Stroke	120	33.73	0.96	0.71–0.99	AQ	Language
Diagnostic Instrument for Mild Aphasia (DIMA)	Clément A. (2022) [139]	French	Healthy population	391	46.5 (19.6)	n.a.	n.a.	n.a.	Language
Dynamic Visual Analogue Mood Scales (D-VAMS)	Barrows P.D. (2018) [140]	English	Stroke	46	63.8 (14.7)	0.9	0.62–0.89	HADS	Language and Cognitive
Ege Aphasia Test (EAT)	Atamaz F. (2007) [141]	Turkish	Healthy population	133	n.a.	n.a.	0.99	n.a.	Language and Cognitive
	Calis F.A. (2013) [142]	Turkish	Stroke, TBI	100	57.7	0.71–0.91	0.99	n.a.	
Frenchay Aphasia Screening Test (FAST)	Enderby P. (1996) [143]	French	Stroke	25	67	n.a.	n.a.	MTDDA FCP	Language
	Paplikar A. (2020) [144]	Indian	Healthy population, Stroke	HP: 116 S: 115	HP: 57.40 (4.12) S: 56.7 (16.9)	n.a.	0.73–0.73	WAB	
Frontal Behavioral Inventory (FBI)	Kertesz A. (2000) [145]	English	FTD	108	67.7	0.89	n.a.	n.a.	Quality of Life
Frontotemporal Dementia Rating Scale (FTD-FRS)	Lima-Silva T.B. (2013) [146]	Portuguese	FTD, Alzheimer	FTD = 12 A = 11	FTD = 66.17 (8.08) A = 67.73 (8.08(	n.a.	n.a.	n.a.	Cognitive
	Turró-Garriga O. (2017) [147]	Spanish	FTD, Alzheimer	FTD: 60 A: 22	FTD: 68.3 (11.7) A: 75.6 (10)	0.89	n.a.	n.a.	
	Lima-Silva T.B. (2018) [148]	Portuguese and English	FTD	97	58.83	0.97	0.97	CDR-FTLD	
Functional Communication Scale (FCS)	Drummond S.S. (2004) [149]	English	TBI, Stroke, Encephalopathy, Brain Tumors	30	32	n.a.	n.a.	RLA cognitive-communication severity levels	Language
Functional Numeracy Assessment (FNA)	Ichikowitz K. (2022) [150]	English	Stroke	25	61.1 (10.32)	n.a.	n.a.	TPT Post-stroke SNS aGHNT-6	Language
Functional Outcome Questionnaire for Aphasia (FOQ-A)	Glueckauf R.L. (2003) [151]	English	Stroke	18	62.9	0.94	n.a.	ASHA-FACS WAB	Quality of Life, Caregivers
	Ketterson T.U. (2008) [152]	English	Stroke	91	63.3	0.94–0.97	n.a.	FOQ-A ASHA-FACS CETI WAB BNT CAI SF-36 GDS	
	Mitasova A. (2015) [153]	Czech	Healthy population, Stroke, Alzheimer	HP: 110 S: 38 A: 8	HP: 63 S: 62 A: 81.5	n.a.	n.a.	n.a.	
	Spaccavento S. (2018) [154]	Italian	Stroke	240	66.59	0.98	0.76–1.00	QLQA FIM FAM AAT	
Health Professionals and Aphasia Questionnaire (HPAQ)	Jensen L.R. (2022) [155]	Danish	Healthy population	270	n.a.	0.91	0.86	n.a.	Quality of Life
Hungarian Aphasia Screening Test (HAST)	Zakariás L. (2023) [156]	Hungarian	Healthy population, Stroke	HP: 51 A: 40 S: 26	HP: 60.5 (12.8) A: 62.2 (14) S: 63.6 (12.7)	0.74	n.a.	WAB	Language
La Trobe Communication Questionnaire (LCQ)	Douglas J.M. (2007) [157]	English	TBI	88	37.88	0.85–0.92	n.a.	n.a.	Language and Quality of Life
	Struchen M.A. (2008) [158]	English	TBI	276	35.88 (13.32)	n.a.	n.a.	n.a.	
Language Assessment Test for Aphasia (LATA)	Toǧram B. (2012) [159]	Turkish	Healthy population, Stroke	HP: 282 S: 92	57.3 (14.1)	0.94–0.99	n.a.	n.a.	Language
Lille’s Apathy Rating Scale (LARS)	Fernández-Matarrubia M. (2015) [160]	Spanish	Healthy population, Alzheimer, Frontotemporal Dementia, Dementia	101	74.3 (7.7)	0.81	0.93	NPI	Quality of Life
Language Screening Test (LAST)	Flamand-Roze C. (2011) [161]	French	Stroke	300	62.6	0.88	0.99	BDAE	Language
	Burgeois-Marcotte J. (2015) [162]	French	Stroke	100	60	n.a.	n.a.	n.a.	Language
	Flowers H.L. (2015) [163]	English	Stroke	109	60 (16.1)	n.a.	n.a.	n.a.	
	Koening-Bruhin M. (2016) [164]	German	Stroke	101	71.37	n.a.	0.81–0.96	TT	
	Yang H. (2018) [165]	Chinese	Stroke	261	58.5 (12.2)	0.95–0.96	1	WAB	
	Sun M. (2020) [166]	Chinese	Stroke	296	66.6 (12.5)	0.91	0.95	n.a.	
	Ramos R.D.L. (2023) [167]	Portuguese	Healthy population, Stroke	HP1: 30 HP2: 70 S: 30	HP1: 50.6 (11.5) HP2: 44.4 (16.7) S: 61.8 (12.9)	0.85	0.91	BDAE	
Language-Specific Attention Treatment (L-SAT)	Peach R.K. (2018) [168]	English	Healthy population	HP: 14	HP: 56.2 54.5	n.a.	n.a.	TEA Stroop Test PASAT OANB VOT	Cognitive
Linguistic Communication Measure (LCM)	Kong A.P.H (2009) [26]	Chinese	TBI, Stroke	10	48.3 (12.4)	n.a.	n.a.	n.a.	Language
	Kong A.P.H (2009) [26]	English	Stroke	16	55.18	n.a.	n.a.	n.a.	
Lothian Assessment for screening cognition in Aphasia (LASCA)	Faiz A. (2016) [169]	English	Brain Tumors, Brain Infections, TBI, Dementia	70	68.21	0.21–0.83	n.a.	ACE	Cognitive
Main Concept Analysis (MCA)	Kong A.P.H. (2015) [170]	Chinese	Healthy population, Stroke	62	n.a.	n.a.	n.a.	n.a.	Language
	Yazu H. (2022) [171]	Japanese	Healthy population, Stroke	HP: 60 S: 20	65.5	n.a.	0.51–1.00	n.a.	
Measure of Stroke Environment (MOSE)	Babulal G.M. (2016) [172]	English	Healthy population, Stroke	43	58.7	0.91	n.a.	BDAE SIS NIHSS	Quality of Life
	Wang W. (2022) [173]	Chinese	Stroke	n.a.	n.a.	0.94	0.94	WHODAS 2.0	
Measure of Skills in Supported Conversation & Measure of Participation in Conversation (MSC-MPC scales)	Kagan A. (2004) [94]	English	Healthy population, Stroke	20	n.a.	n.a.	n.a.	n.a.	Language, Caregivers
	Muò R. (2019) [174]	Italian	Healthy population, Stroke	16	61.4 (15.5)	n.a.	0.91–0.98	ASHA-FACS	
Mental Deterioration Battery (MDB)	Carlesimo G.A. (1996) [175]	Italian	Stroke	340	n.a.	n.a.	n.a.	n.a.	Cognitive
Mini Mental State Examination (MMSE)	Vigliecca N.S. (2012) [176]	Spanish	Stroke	109	50.57 (8.13)	0.92–0.95	n.a.	BAE	Cognitive
Mississippi Aphasia Screening Test (MAST)	Nakase-Thompson R. (2005) [177]	English	Healthy population, Stroke	HP: 36 S1: 38 S2: 20	HP: 46.6 (19.2) S1: 61.7 (12.7) S2: 58.7 (15.7)	n.a.	n.a.	n.a.	Language
	Kostálová M. (2008) [178]	Czech	Stroke	194	68	n.a.	n.a.	WAB	
	Romero M. (2012) [179]	Spanish	Stroke	29	55.9 (12.9)	n.a.	n.a.	BDAE	
	Khatoonabadi A.R. (2015) [180]	Persian	Stroke	40	52.3 (8.2)	0.64	0.96	n.a.	
	Nursi A. (2019) [181]	Estonian	Stroke	176	72.5	0.88–0.95	n.a.	n.a.	
Montreal Cognitive Assessment (MoCA)	Lim P.A. (2016) [182]	English	Stroke, TBI	40	56.3	n.a.	n.a.	n.a.	Cognitive
Montreal Evaluation of Communication (MEC)	Le Dorze G. (2010) [183]	French	Healthy population, Stroke, Dementia	S: 15 D: 16 HP: 62	S: 73.13 D: 82.5 HP: n.a.	n.a.	n.a.	REFCP	Healthcare professionals
Montreal Evaluation of Communication Brief Battery (MEC-B)	Casarin F.S. (2019) [184]	Portuguese	Healthy population	HP: 324	HP: 44.57 (15.57)	0.70	0.75–1.00	n.a.	
Montreal-Toulouse Language Assessment Battery (MTL-BR)	Pagliarin K.C. (2014) [185]	Portuguese	Healthy population, Stroke	HP: 463 S: 74	HP: 44.84 (15.12) S: 58.98 (1.10)	0.79–0.90	n.a.	MAC Brief MAC WAIS III NEUPSLIN	Language
	Pagliarin K.C. (2015) [186]	Portuguese	Healthy population, Stroke	HP: 25 S: 104	HP: 56.6 (12.5) S: 59.19 (1.24)	n.a.	n.a.	n.a.	
Brief Montreal-Toulouse Language Assessment Battery (MTL-BR)	Altmann R.F. (2020) [187]	Portuguese	TBI, Infections, Brain Tumor	65	42.55	n.a.	n.a.	n.a.	Language
Multiple-Choice Test of Auditory Comprehension for Aphasia (MCTAC)	Hallowell B. (2009) [188]	Russian	Stroke, TBI, Encephalitis	75	47.4	n.a.	n.a.	n.a.	Language
Multilingual Aphasia Examination (MAE)	Rey G.J. (2001) [189]	Spanish	TBI	40	42.7	n.a.	n.a.	n.a.	Cognitive
Multilingual Aphasia Examination—Visual Naming Test (MAE-VNT)	Axelrod B.N. (1994) [190]	English	Mental disorders	100	52 (15)	n.a.	n.a.	n.a.	
Naming and Oral Reading for Language in Aphasia 6-Point scale (NORLA-6)	Pitts L.L. (2018) [191]	English	Stroke	91	54.8 (11.9)	n.a.	0.90–0.92	GORT-4 BNT	Language
Naming Assessment in Multicultural Europe (NAME)	Franzen S. (2023) [192]	Dutch and Turkish	Alzheimer, Dementia	176	67	n.a.	n.a.	RUDAS	Language
Exploration of Natural Metalinguistic Skills in Aphasia (MetAphAs)	Mac-Kay A.P.M.G. (2020) [193]	Portuguese	Healthy population	72	n.a.	0.91	n.a.	n.a.	Language
Neuro-Cognitive Assessment Battery for Stroke Patients (NCABS)	Mahmood S.N. (2018) [194]	English	Stroke	121	70	0.92	n.a.	MoCA	Cognitive
Neuropsychiatric Inventory (NPI)	Yiannopoulou K.G. (2019) [195]	Greek	FTD	311	64 (8)	n.a.	n.a.	n.a.	FTD
Non-language based Cognitive Assessment (NLCA)	Wu J.B. (2017) [196]	Chinese	Healthy population, Stroke, Mild Cognivite Impairment	157	n.a.	0.81	n.a.	MoCA	Cognitive
Northwestern Anagram Test (NAT)	Weintraub S. (2009) [197]	English	Dementia	16	60	n.a.	n.a.	NAVS WAB PPVT-IV PPT	Cognitive
Northwestern Assessment of Verbs and Sentences (NAVS)	Thompson C. (2012) [198]	English	n.a.	n.a.	n.a.	n.a.	n.a.	n.a.	Language
	Cho-Reyes S. (2012) [199]	English	Stroke	59	n.a.	n.a.	n.a.	n.a.	
	Wang H. (2016) [200]	Chinese	Healthy population, Stroke	HP: 15 S: 15	HP: 59 S: 56	n.a.	n.a.	n.a.	
Northwestern Assessment of Verbs and Sentences—Verb Naming Test (NAVS-VNT) subtest	Fergadiotis G. (2023) [201]	English	Stroke	107	61.52	n.a.	n.a.	n.a.	
Northwestern Narrative Language Analysis (NNLA)	Hsu C.J. (2017) [129]	English	Healthy population, Stroke	18	HP: 58.2 S: 57.4	n.a.	n.a.	n.a.	Language
Object and Action Naming Battery: Objects (OANBObj) and Object and Action Naming Battery: Actions (OANBAct)	Peach R.K. (2018) [168]	English	n.a	20	56.9 (4.0)	n.a.	n.a.	n.a.	Language
Oxford Cognitive Screen (OCS)	Demeyere N. (2015) [202]	English	Stroke	208	71.1	n.a.	0.33–0.77	MoCA CAT BDAE BIT BCoS Weechsler’s test	Cognitive
	Kong A.P. (2016) [203]	Chinese	Healthy population, Stroke	116	55.2	0.73	n.a.	WAB MMSE MoCA Albert’s test	
	Ramos C.C.F. (2018) [204]	Portuguese	Healthy population	30	61.2 (6.31)	n.a.	n.a.	n.a.	
	Valera-Gran D. (2018) [205]	Spanish	Healthy population, Stroke	HP: 54 S: 57	n.a.	0.91	0.17–0.79	MoCA Barcelona test BI	
	Hong W. (2018) [206]	Chinese	Healthy population, Stroke	HP1: 60 HP2: 60 S: 100	HP1: 29 (3.4) HP2: 58.7 (6.6) S: 59.2 (8.8)	0.30–0.52	0.33–0.77	MoCA	
	Shendyapina M. (2018) [207]	Russian	Healthy population, Stroke	HP: 60 S: 205	HP: 61 (19.03) S; 62 (15.78)	n.a.	n.a.	n.a.	
	Huygelier H. (2022) [208]	Dutch	Stroke	193	65	n.a.	0.47–0.96	MoCA	
	Webb S.S. (2022) [209]	English	Stroke	347	73 (13.36)	n.a.	n.a.	CLQT RBANS BIT	
	Bormann T. (2023) [210]	German	Healthy population	100	71.2 (8.9)	n.a.	0.46–0.87	MoCA	
	Murphy D. (2023) [211]	English	Stroke	316	73.1 (12.76)	n.a.	n.a.	n.a.	
	Cho E. (2024) [212]	Korean	Healthy population	97	54.3 (9.7)	n.a.	n.a.	n.a.	
Paced Auditory Serial Addition Test (PASAT)	Nikravesh M. (2017) [213]	Iranian	Stroke	25	49.52	n.a.	0.95	WAIS-R	Cognitive
Pain Assessment Checklist for Seniors with Limited Ability to Communicate (PACSLAC)	Fuchs-Lacelle S.M.A. (2004) [214]	English	Caregivers	40	49 (13.0)	0.92	n.a.	n.a.	Quality of Life
	Aubin M. (2007) [215]	French	Dementia	G1: 86 G2: 26	G1: 84 (6.4) G2: 75.8 (15.3)	n.a.	n.a.	n.a.	
	Cheung G. (2008) [216]	English	Dementia	50	82.9 (7.2)	n.a.	n.a.	n.a.	
	Kim E.K. (2014) [217]	Korean	Dementia	307	80.72 (0.45)	n.a.	n.a.	n.a.	
	Chan S. (2014) [218]	English	Alzheimer, Dementia, Other	124	83.94 (7.95)	n.a.	n.a.	n.a.	
	Thé K.B. (2016) [29]	Portuguese	Alzheimer, Dementia	50	87.8 (6.5)	0.83	0.64–0.85	n.a.	
	Büyükturan Ö. (2018) [219]	Turkish	Dementia	112	70.12 (5.94)	n.a.	n.a.	n.a.	
	Haghi M. (2020) [220]	Iranian	Dementia	138	74.5 (8.9)	0.72–0.84	0.76	GDS	
	De Vries N.J. (2023) [221]	Dutch	Stroke	60	79.3	0.33–0.86	0.07–0.88	NRS/VAS FPS	
Perceived Stress Scale (PSS)	Pompon R.H. (2018) [222]	English	Stroke	74	64.53	n.a.	n.a.	VAS	Quality of Life
Philadelphia Brief Assessment of Cognition (PBAC)	Libon D.J. (2011) [223]	English	Healthy population, Alzheimer, Frontotemporal Dementia, Corticobasal Syndrome, Other	270	64	0.59–0.76	n.a.	MMSE	Cognitive
Philadelphia Naming Test (PNT)	Walker G.M. (2012) [224]	English	Stroke	86	58	0.95–0.96	n.a.	BNT	Language
	Swiderski A.M. (2023) [225]	English	Stroke	24	64.50 (13.87)				
Picture Naming Test (PNT)	Tahanzadeh B. (2017) [226]	Farsi (Iran)	Healthy Population, Stroke	HP: 30 S: 10	HP: 47.97 (9.48) S: 47.2 (10.58)	0.92–0.95	0.97–0.98	n.a.	Language
	Macoir J. (2021) [227]	English	Healthy population, Mild Cognitive Impairments, Alzheimer, Stroke	227	68	0.81	n.a.	BNT	
	Vivas L. (2022) [228]	Spanish	Healthy population, Brain damage, Degenerative neurological diseases	148	61.18	n.a.	n.a.	n.a.	
	Khatibani M.N. (2022) [229]	Persian	Healthy population, Alzheimer	HP: 50 A: 20	HP: 74 (5.9) A: 74.4 (5.8)	n.a.	n.a.	n.a.	
Preliminary Neuropsychological Battery (PNB)	Hobson J.P. (2003) [230]	English	Stroke	271	72.8	0.89	n.a.	CAMCOG-R	Cognitive
Progressive Aphasia Language Scale (PALS)	Jafari S. (2018) [231]	English and Iranian	Dementia	30	61.7 (7.2)	n.a.	0.81–1.00	n.a.	Language
Progressive Aphasia Severity Scale (PASS)	Sapolsky D. (2010) [232]	English	Dementia	40	n.a.	n.a.	0.91–1.00	WAB BDAE PASS CSB	Language
	Petrillo S.M. (2022) [233]	Italian	Dementia	n.a.	n.a.	n.a.	n.a.	n.a.	
Prueba Argentina Psicolingüística de Denominación de Imágenes (PAPDI)	Manoiloff L. (2018) [234]	Spanish	Healthy population, Alzheimer, TBI	177	n.a.	n.a.	n.a.	BNT	Language
Psycholinguistic Assessments of Language Processing in Aphasia (PALPA)	Pinto-Grau M. (2021) [235]	English	Healthy population	100	64.0 (10.4)	0.78	0.62–0.93	n.a.	Language
Pyramids and Palm Trees Test (PPT)	Klein L.A. (2009) [28]	English	Healthy population	90	19.8	n.a.	n.a.	TMT	Cognitive
Quality of life questionnaire for aphasics (QLQA)	Spaccavento S. (2014) [5]	Italian	Stroke	164	68.39 (10.29)	0.96	0.65–0.98	FIM FAM	Quality of Life
Quick aphasia battery (QAB)	Wilson S.M. (2018) [236]	English	Stroke	73	63.5 (17.3)	n.a.	0.98–0.99	WAB	Language
	Zhu D. (2023) [237]	Chinese	Stroke	128	63.2	0.96	0.99	WAB	
	Parlak M.M. (2024) [238]	Turkish	Healthy population, Stroke	188	64.26 (10.21)	n.a.	n.a.	LATA	
Reading Comprehension Battery in Aphasia (RCBA)	Van Demark A.A. (1982) [239]	English	Stroke, TBI	36	44.2	n.a.	n.a.	GSRT PICA	Language
	Flanagan J.L. (1997) [77]	English	Healthy population	31	63.74 (7.4)	n.a.	n.a.	n.a.	
Repeatable Battery for the Assessment of Neuropsychological Status (RBANS)	Wilde M.C. (2006) [240]	English	Stroke	210	61.91	n.a.	n.a.	MMSE VFD COWAT BDAE	Language
	Merz Z.C. (2018) [241]	English	TBI, Stroke	2057	62.44	n.a.	n.a.	COWA MAE BNT	
Russian Aphasia Test (RAT)	Ivanova M.V. (2021) [242]	English, Dutch, Russian	Stroke, TBI, Brain Tumors, Other	85	57.6	0.79–0.98	0.83–0.99	GAQ ASA	Language
Scandinavian Stroke Scale (SSS)	Thommessen B. (2002) [243]	Norwegian	Stroke	32	75.5	n.a.	n.a.	n.a.	Language
Scenario Test (ST)	Hilari K (2018) [244]	English	Stroke	94	60.9	0.96	0.86–1.00	FAST ASHA-FACS CLQT CAT BUPS LAS	Language
	van der Meulen I. (2010) [245]	Dutch	Stroke	147	58.4	0.96	0.98	ANELT	
	Charalambous M. (2022) [246]	Greek	Stroke	54	54.3	0.95	0.99	BDAE CETI PCRM AIQ-21 SAQOL-39	
	Kodani Y. (2024) [247]	Japanese	Healthy population, Stroke	HP: 27 S: 34	HP: 65.56 (14.73) S: 63.82 (13.02)	0.93	0.95–1.00	n.a.	
ScreeLing (SL)	El Hachioui H. (2012) [248]	Dutch	Stroke	302	66.61	0.93–0.95	n.a.	TT SSR	Language
Screening Aphasia Test (SAT)	Abou-Elsaadd T. (2018) [249]	Arabic	Stroke	30	57	0.86–0.93	n.a.	CAT	Language
	Banco E. (2023) [250]	Italian	Healthy population, Stroke	HP: 329 S: 139	HP: 51.12 (17.76) S: 63.57 (14.07)	n.a.	n.a.	n.a.	
Screening Léxico para las Afasias (SLA)	Peña-Chávez R. (2014) [251]	Spanish	Healthy population, Stroke	HP: 29 S: 29	n.a.	0.87	n.a.	BETA	Language
Screening for Aphasia in NeuroDegeneration (SAND)	Catricalà E. (2018) [252]	Italian	Healthy population	134	63.28 (11.19)	n.a.	n.a.	n.a.	Language
	Battista P. (2018) [253]	Italian	Dementia	205	63.30 (11.30)	0.86	n.a.	n.a.	
	Picillo M. (2019) [254]	Italian	Progressive Supranuclear Palsy (PSP), Parkinsonism, Parkinson Disease	127	71	0.88	n.a.	CaGi naming ENPA	
Screening instrument for neuropsychological impairment in stroke (SINS)	Sødring K.M. (1998) [255]	Norwegian	Stroke	87	74	0.82–0.90	n.a.	ASB SINS	Cognitive
Screening Test for Aphasia and Dysarthria (STAD)	Araki K. (2022) [256]	Japanese	Stroke, TBI	314	72.7 (12.9)	n.a.	n.a.	WAB AMSD	Language, Cognitive
Sentence Repetition Test (SRT)	Spreen O. (1963) [257]	English	Stroke	n.a.	n.a.	n.a.	n.a.	WAIS-III	Language
	Meyers J.E. (2000) [258]	English	Healthy population	104	39.98	n.a.	n.a.	n.a.	
Severity-Calibrated Aphasia Naming Test (SCANT)	Walker G.M. (2022) [259]	English	Stroke	183	58	0.99	n.a.	WAB PNT	Language
Shewan Spontaneous Language Analysis (SSLA)	Shewan C.M. (1988) [260]	English	Healthy Population, Stroke	HP: 30 S: 47	n.a.	n.a.	n.a.	n.a.	Language
Short Aphasia Test for Gulf Arabic speaking populations (SATG)	Altaib M.K. (2021) [261]	Arabic	Healthy Population, Stroke	HP: 37 S: 31	38.40 (12.23)	n.a.	0.94–1.00	n.a.	Language
Signs of Depression Scale (SODS)	Bennett H.E. (2006) [262]	English	Stroke	100	n.a.	−0.04–0.84	n.a.	SADQ-H VAMS VASES	Cognitive
	van Dijk M.J. (2017) [263]	Dutch	Stroke	116	70 (14.6)	0.69	0.66–0.80	CIDI	
	van Dijk M.J. (2018) [264]	Dutch	Stroke	58	59.3	0.71	0.79	PHQ-9	
Sickness impact profile (SIP-65)	Bénaim C. (2003) [265]	French	Stroke, TBI, Meningioma	79	55	n.a.	n.a.	SIP-136	Quality of Life
	van Dijk M.J. (2018) [264]	Dutch	Stroke	58	59.3	n.a.	n.a.	n.a.	
Simple Aphasia Stress Scale (SASS)	Laures-Gore J. (2019) [266]	English	Stroke	33	n.a.	n.a.	0.70	n.a.	Quality of Life
Size/Weight Attribute Test (SWAT)	Yoo Y. (2016) [267]	Korea	Frontotemporal Dementia	95	72.96	n.a.	0.83	n.a.	Language
Social Activities Checklist (SOCACT)	Aujla S. (2015) [112]	English	Healthy population, Stroke	HP: 75 S: 31	HP: 74 S: 71	−0.25–0.55	n.a.	n.a.	Quality of Life
Standard Language Test of Aphasia (SLTA)	Tsutsumiuchi K. (2020) [268]	Japanese	Stroke	20	68.5 (12.5)	n.a.	n.a.	WAB	Language
Story Retell Procedure (SRP)	Doyle P.J. (2001) [269]	English	Stroke	15	n.a.	n.a.	n.a.	n.a.	Language
	McNeil M.R. (2002) [270]	English	Healthy population, Stroke	HP: 15 S: 31	HP: 43.7 (17.2) S: 62.7 (9.1)	n.a.	n.a.	n.a.	
	Hula W.D. (2003) [271]	English	Healthy population, Stroke	HP: 31 S: 15	n.a.	n.a.	n.a.	n.a.	
	McNeil M.R. (2002) [270]	English	Stroke	13	61.54 (11.75)	n.a.	n.a.	n.a.	
	Ortiz K.Z. (2024) [272]	Portuguese	Healthy population, Stroke	HP: 14 S: 7	n.a.	n.a.	n.a.	n.a.	
Stroke and Aphasia Quality of Life Scale–39 (SAQOL-39)	Hilari K. (2003) [273]	English	Stroke	83	61.67	0.93	0.98	SAQOL	Quality of Life
	Posteraro L. (2004) [274]	Italian	Stroke	43	65.4	0.94	n.a.	n.a.	
	Hilari K. (2007) [275]	English	Stroke	50	63.4	n.a.	0.72–0.96	n.a.	
	Hilari K. (2009) [276]	English	Stroke	96	69.7	0.92–0.95	0.92–0.98	n.a.	
	Lata-Caneda M.C. (2009) [277]	Spanish	Stroke	23	57 (10.64)	n.a.	n.a.	n.a.	
	Efstratiadou E.A. (2012) [278]	Greek	Stroke	60	66.68	0.96	0.83–0.99	BI FAI GHQ-12 FAST MoCA	
	Kiran S. (2013) [279]	Indian	Stroke	32	n.a.	0.90	0.80	n.a.	
	Mitra I.H. (2015) [280]	Indian	Stroke	87	69.7 (14.1)	0.98	0.90	n.a.	
	Kamiya A. (2015) [281]	Japanese	Stroke	54	66.4	0.74–0.90	0.89–0.97	n.a.	
	Calis A.F. (2016) [282]	Turkish	Stroke	62	55.5 (12.8)	0.80–0.89	0.70–0.97	BRS	
	Guo Y.E. (2016) [283]	Malaysian/English	Stroke	97	83.2	0.97	0.99	EQ5D	
	Noyan-ErbaŞ A. (2016) [284]	Turkish	Stroke	30	50.8 (10.5)	0.70–0.97	0.97	n.a.	
	Van Ewijk L. (2017) [285]	Dutch	Stroke	13	62	0.84–0.91	0.70–0.93	VAS	
	Qiu W. (2019) [286]	Chinese	Stroke	84	55.26	n.a.	n.a.	n.a.	
	Van Ewijk L. (2019) [287]	Dutch	Stroke	141	60.4	0.96	n.a.	EQ5D	
	Kariyawasam P.N. (2020) [288]	Singalese	Stroke	115	67.07	0.98	0.92	n.a.	
	Kristinsson S. (2021) [289]	Icelandic	Stroke	30	54.6	0.94	0.95	n.a.	
	Azizbeigi-Boukani J. (2021) [290]	Iranian	Stroke	30	59.40	0.86–0.95	0.76–0.97	SF-12 BI VAS	
	Vuković M. (2022) [291]	Serbian and English	Stroke	90	56.3	0.91–0.97	0.87–0.94	n.a.	
	Sommer J.B. (2024) [292]	Danish	Stroke	72	60 (12.7)	n.a.	n.a.	n.a.	
Stroke Aphasic Depression Questionnaire (SADQ)	Sutcliffe L.M. (1998) [293]	English	Stroke	70	72.4	0.82	n.a.	HADS Wakefield Depression Inventory	Quality of Life
	Sackley C.M. (2006) [294]	English	Stroke	82	n.a.	n.a.	n.a.	HADS	
	Hacker V.L. (2010) [295]	English	Stroke	125	73 (13)	0.68	n.a.	BASDEC	
	Cobley C.S. (2012) [296]	English	Stroke	165	68.6	0.77	n.a.	VAMS VASES	
Stroke Communication Scale (ICF-SCS)	Batista Dallaqua G. (2019) [297]	Portuguese	Healthy population, Stroke	HP: 22	HP: 49	n.a.	n.a.	n.a.	Caregivers
Stroke Social Network Scale (SSNS)	Northcott S. (2013) [298]	English	Stroke	87	69.7	0.85	n.a.	SAQOL-39 FAST	Quality of Life
Stroke Specific Quality of Life Scale (SS-QOL)	Hilari K. (2001) [299]	Dutch	Stroke	80	n.a.	n.a.	n.a.	n.a.	Quality of Life
Sydney Language Battery (SYDBAT)	Savage S. (2013) [300]	English	Dementia	111	65.7 (7.8)	n.a.	n.a.	n.a.	Language
	Janseen N. (2022) [301]	English and Dutch	Alzheimer	45	67.2	n.a.	n.a.	n.a.	
TEst Français de RÉpétition de Phrases (TEFREP)	Bourgeois-Marcotte J. (2015) [302]	French	Stroke, TBI, Dementia	80	62.6	0.84	n.a.	n.a.	Language
Test de dénomination d’actions par vidéos (T-DAV)	Spigarelli M. (2022) [303]	French	Alzheimer	64	79.69 (6.68)	0.77	n.a.	DVL-38	Language
	Macoir J. (2023) [304]	French	Healthy population, Mild Cognitive Impairment	30	n.a.	n.a.	n.a.	n.a.	
Test de Dénomination de Québec-30 images (TDQ-30)	Macoir J. (2019) [227]	French	Healthy population, Mild Cognitive Impairment, Alzheimer, Stroke	HP: 277 MCI: 14 A: 10 S: 9	HP: 68 (9.3) MCI: 74.5 (6.7) A: 76.1 (4.6) S: 62.8 (10.5)	0.81	n.a.	BNT	Language
Token Test (TT)	Park G.H. (2000) [305]	English	Stroke	12	70.08	n.a.	0.71–0.99	n.a.	Language and Cognitive
	Hula W. (2006) [306]	English	Stroke	30	57.7	n.a.	n.a.	n.a.	
	Bakhtiar M. (2020) [307]	Chinese	Healthy population, Stroke	74	58.96	n.a.	0.86–0.96	WAB	
	McNeil M.R. (2015) [308]	Korean and English	Healthy population, Stroke	60	63	n.a.	n.a.	PICA	
	Paci M. (2015) [309]	Italian	Stroke	24	78.8	n.a.	0.12–1.00	n.a.	
	Bakhtiar M. (2020) [307]	Chinese	Healthy population, Stroke	HP: 42 S: 32	HP: 58.65 (7.94) S: 58.69 (6.25)	n.a.	0.86–0.96	WAB	
Verb and Sentence Test (VAST)	Bastiaanse R. (2003) [310]	Dutch and English	Healthy population, Stroke	104	68.40	n.a.	n.a.	n.a.	Language
Visual Analog Mood Scales (VAMS)	Temple R.O. (2004) [311]	English	Dementia	31	77.5 (8.5)	n.a.	n.a.	POMS	Quality of Life
	Kontou E. (2012) [312]	English	Healthy population, Stroke	121	69	0.74–0.80	n.a.	HADS VASES SADQH-21	
	Barrows P.D. (2018) [313]	English	Stroke	46	63.8 (14.7)	0.95	n.a.	HADS	
Visual Analog Mood Scales—Revised (VAMS-R)	Kontou E. (2012) [312]	English	Healthy population, Stroke	HP: 50 S: 71	HP: 66.7 (9.7) S: 69 (12.33)	0.74–0.80	n.a.	HADS VASES SADQH-21	
Visual Analogue Self-Esteem Scale (VASES)	Brumfitt S.M. (1999) [314]	English	Healthy population, Stroke	HP: 243 S: 34	HP: n.a. S: 67.05	0.86	n.a.	RSE GHQ HADS VAMS	Quality of Life
	Bennett H.E. (2006) [262]	English	Stroke	100	n.a.	0.83–0.85	n.a.	SADQ-H SODS VAMS	
Visual-Analogue Test Assessing Anosognosia for Language Impairment (VATA-L)	Cocchini G. (2010) [315]	English	Stroke, TBI	65	57	n.a.	n.a.	AAT	Anosognosia
Western Aphasia Battery (WAB)	Shewan C.M. (1980) [316]	English	Stroke	n.a.	n.a.	n.a.	n.a.	NCCEA	Language
	Kim H. (2004) [317]	Korean	Healthy population	462	57.1	n.a.	n.a.	BNT	
	Bakheit A.M.O. (2005) [318]	English	Stroke	67	71.9	n.a.	n.a.	CETI	
	Hula W. (2010) [319]	English	Stroke	108	63.1	n.a.	n.a.	n.a.	
	Neves Mde B. (2014) [320]	Portuguese	TBI, Brain Tumors, Brain Inflammations, Other	30	n.a.	n.a.	n.a.	n.a.	
	Boucher J. (2022) [321]	French	Healthy population	62	70.95				
Western Aphasia Battery—Revised (WAB-R)	Nilipour R. (2014) [322]	Persian	Stroke, TBI	100	51.95 (8.59)	0.71–0.91	n.a.	n.a.	
	Dekhtyar M. (2020) [323]	English	TBI, Stroke	20	55	n.a.	0.99	n.a.	
	El Ouardi L. (2023) [324]	Arabic	Stroke	52	56.6	0.78–0.95	0.96–0.99	n.a.	
	Rao L.A. (2022) [325]	English	Dementia	19	64.4	n.a.	n.a.	n.a.	
Western Aphasia Battery—Language Quotient (WAB-LQ)	Shewan C.M. (1986) [326]	English	Stroke	94	65.02	0.91	n.a.	n.a.	
Western Aphasia Battery—Revised (WAB-R) Picture Description Task	Marcotte K. (2024) [327]	French	Healthy population	66	n.a.	n.a.	n.a.	n.a.	
World Retrieval in Aphasic Discourse (WRAD)	Boyle M. (2013) [328]	English	Stroke, TBI	12	62 (13.5)	n.a.	n.a.	n.a.	Language

SD: Standard Deviation; n: number; n.a.: not available; TBI: Traumatic Brain Injury; HP = Healthy Population; FTD = Frontotemporal Degeneration; PS: Post-stroke; PPA: Primary Progressive Aphasia; PPAOS: Primary progressive apraxia of speech; BDAE-SF = Boston Diagnostic Aphasia Examination 3rd Edition—Short Form; WAI = Working Alliance Inventory; ADRS = Aphasia Depression Rating Scale; SAQOL = Stroke and Aphasia Quality of Life Scale; AAT = Aachener Aphasia Test; HADS = Hospital Anxiety and Depression Scale; WAB-R = Western Aphasia Battery—Revised; TEA = Test of Everyday Attention; PASAT = Paced Auditory Serial Addition Test; OANB = Object and Action Naming Battery; VOT = Hooper Visual Organization Test; CETI = Communicative Effectiveness Index; ASSIDS = Assessment of Intelligibility of Dysarthric Speech; QLQA = Quality of Life Questionnaire for Aphasics; FIM = Functional Independence Measure; FAM = Functional Assessment Measure; VAS = Visual Analogue Scale; WAIS III = Weschler Adult Intelligence Scales; NEUPSLIN = Brazilian Brief Neuropsychological Assessment Battery; BOSS CAPD = Communication Associated Psychological Distress Scale of the Burden of Stroke Scale; TROG-H = Test for the Reception of Grammar—Hungarian; BRS = Brunnstrom Recovery Study; I-ASHA-FACS = American Speech–Language and Hearing Association—Functional Assessment of Communication Skills for Adults; CDR-FTLD = Clinical Dementia Rating Scale for Frontotemporal Lobar Degeneration; ADD = Aphasia Language Assessment Test; MTDDA = Minnesota Test for Differential Diagnosis of Aphasia; FCP = Functional Communication Profile; MPC = Measure of Participation in Conversation; ASR = Aphasia Severity Rating; CLQT = Cognitive Linguistic Quick Test; BUPS = Birmingham University Praxis Screen; LAS = Limb Apraxia Screen; DVL = Denomination de Verbes Lexicaux en images Test; BECLA = Batterie d’Evaluation Cognitive du Langage; MT-86 = Protocole Montréal–Toulouse d’examen linguistique de l’aphasie; MEC = Protocole Montre ´al d’E ´valuation de la Communication; WMS-IV = Wechsler Memory Scale—IV; PPTT = Pyramids and Palm Trees Test; PICA = Porch Index of Communicative Ability; COPE = Carers of Older People in Europe; NANDA-I = Nursing Assessment of Ability to Communicate among Patients with Aphasia questionnaire; NOC = Nursing Outcomes Classification; NPI = Neuropsychiatric Inventory; CSB = Cambridge Semantic Battery; NAART = North American Adult Reading Test; RUDAS = Rowland Universal Dementia Assessment Scale; BIT = Behavioral Inattention Test; ENPA = Esame Neuropsicologico dell’Afasia; TPT = Time Pressure Task; SNS = Subjective Numeracy Scale; aGHNT-6 = Aphasia-friendly General Health Numeracy Test; RLA = Ranchos Los Amigos; PPVT = Peabody Picture Vocabulary Test; FCP = Functional Communication Profile; BI = Barthel Index; FAI = Frenchay Activities Index; GHQ-12 = 12-item General Health Questionnaire; LCF = Levels of Cognitive Functions scale; CADL = Communication Activities of Daily Living; SF-12 = 12-Item Short-Form Health Survey; GAQ = General Aphasia Quotient; ASA = Assessment of Speech in Aphasia; NRS = Numeric Rating Scale; FPS = Faces Pain Scale; PCRM = Ravens Color Progressive Matrices; VRS = Verbal Rating Scale; SSR = Spontaneous Speech Rating; PHQ-9 = Patient Health Questionnaire—9; GORT-4 = Gray Oral Reading Test—Fourth Edition; SIS = Stroke Impact Scale; NIHSS = National Institute of Health Stroke Scale; GAD-7 = Generalised Anxiety Disorder—7; CAI = Caregiver Appraisal Inventory; SF-36 = Short-Form Health Survey 36; GDS = Geriatric Depression Scale; REFCP = Revised Edinburgh Functional; NAI = Neurolinguistic Abilities Index; NCCEA = Neurosensory Center Comprehensive Examination for Aphasia; RSE = Rosenberg Self-Esteem Scale; TMT = Trail Making Test; GSRT = Gates Silent Reading Test; COOP-WONCA = Dartmouth Coop Functional Health Assessment Charts—World Organisation of Family Doctors; CAMCOG-R = Cambridge Cognitive Examination—Revised; VDS = Verbal Descriptor Scale; CNPI = Checklist of Nonverbal Pain Indicators; FTDFRS = Frontotemporal Dementia Functional Rating Scale; ADAS = Alzheimer Disease Assessment Scale; CIDI = Composite International Diagnostic Interview; POMS = Profile of Mood States; BASDEC = Brief Assessment Schedule Depression Cards.

As regards language assessment, the most widely used instrument is the Language Screening Test (LAST), validated in French [164,165], English [166], German [167], Chinese [168,169], and Portuguese [170]. In the context of the quality of life of the patient with aphasia, the most widely translated tool is the Stroke and Aphasia Quality of Life Scale—39 (SAQOL-39), validated in English [281,283,284], Italian [282], Spanish [285], Greek [286], Indian [287,288], Japanese [289], Turkish [290,292], Malaysian [291], Dutch [293,295], Chinese [294], Singalese [296], Icelandic [297], Iranian [298], and Serbian [299]; the most translated tool in the cognitive domain is the Oxford Cognitive Screen (OCS), validated in English [208,215,217], Chinese [209,212], Portuguese [210], Spanish [211], Russian [213], Dutch [214], and German [216]; and, among the tools aimed at caregivers, the most used is the Functional Outcome Questionnaire for Aphasia (FOQ-A), validated in English [154,155], Czech [156], and Italian [157]. The Aphasia Rapid Test (ART), validated in French [63], Italian [64], Russian [65], and Turkish [66], was the only identified tool in the acute stroke assessment category; the Visual-Analogue Test Assessing Anosognosia for Language Impairment (VATA-L), validated in English [323], was the only instrument in the anosognosia domain; the Auditory–Perceptual Rating of Connected Speech in Aphasia (APROCSA), validated in English [78], for the auditory and perceptual assessment category; and the City Gesture Checklist (CGC), validated in English [106], for the praxis assessment category. As for multidimensional assessment tools, the most used is the Token Test, validated in English [312,313], Chinese [312], Korean [314], and Italian [262], which evaluates the linguistic and cognitive aspect.

### 3.3. Study Population

Most of the studies (n = 104) presented post-stroke aphasia patients as the population of interest. Other represented populations are healthy subjects (n = 52) and dementia patients (n = 18).

### 3.4. Risk of Bias Within Studies

The risk of bias of the 238 included studies was variable. The methodological quality was analyzed using the COSMIN checklist [23]; the scores obtained by individual studies are reported in Table 3. In general, the studies were found to have a fairly good quality. Item 1 (PROM development), Item 2 (Content validity), Item 5 (Cross-cultural validity/Measurement invariance), Item 6 (Reliability), and Item 8 (Criterion validity) had the highest number of positive results. The less represented Items were 4 (Internal consistency), 7 (Measurement error), 9 (Hypotheses testing for construct validity), and 10 (Responsiveness). The study with the highest number of positive results in the COSMIN checklist is from Kavakci et al. (2022), validating the Aphasia Rapid Test (ART) [66], while the study with the lowest number of positive results is that of Spezzano et al. (2013), validating An Object and Action Naming Battery (An O&A Battery) [55].

## 4. Discussion

Assessing aphasia in neurological patients is a critical aspect of clinical management, as it significantly impacts communication and overall quality of life [81]. The early and accurate identification of the type and severity of aphasia enables clinicians to tailor rehabilitation interventions, enhancing the effectiveness of treatment; moreover, it helps predict potential functional, social, and psychological outcomes, supporting a multidisciplinary approach to treatment that involves the patient, caregivers, and family members [329,330]. Therefore, the aim of this review was to identify currently available and validated measurement tools (such as scales, tests, and questionnaires) that assess various aspects of communication and cognition in individuals with aphasia, providing an analysis of their validity and psychometric properties.

A systematic review of the existing scientific literature was carried out up to August 2024, allowing the identification of 238 studies reporting the psychometric properties of aphasia assessment tools. Following the Consensus-based Standards for the selection of health Measurement Instruments (COSMIN) checklist [23], authors have highlighted the psychometric properties of 181 assessment tools for the assessment of PWA. No time limits were set so as not to exclude any study that could have made important contributions to the review. Existing studies were published between 1963 [265] and 2024 [216]. The systematic review revealed significant heterogeneity among the assessment tools, with 52.5% of the 181 tools having only a single development or validation article. While it is beneficial to investigate the various domains of aphasia—given its impact on multiple aspects of language and cognition—this diversity poses a scientific and methodological challenge: the lack of uniformity in measurement tools hampers the comparability of clinical studies and limits the possibility of carrying out meta-analysis studies. Additionally, many of these instruments do not demonstrate adequate methodological quality, further limiting their reliability and validity in clinical and research settings. According to the COSMIN checklist, ensuring the validity of an assessment tool requires the transparent and thorough documentation of its psychometric properties; while having a single development or validation article can provide initial evidence, it may not suffice to establish a comprehensive evaluation of the tool’s validity across different contexts and populations. Manuals can be valuable sources, but peer-reviewed articles are often necessary in order to provide the additional and independent verification of psychometric properties.

Regarding the psychometric properties analyzed according to the COSMIN checklist, the most represented items concern the development of the outcome measure and the analysis of its content validity, cross-cultural validity, reliability, and criterion validity. Item 5 (cross-cultural validity) is among the most represented: evaluating cross-cultural validity in language assessment scales is indeed crucial as language and communication are deeply influenced by cultural and linguistic contexts [331,332]. This psychometric property makes it possible to reduce cultural biases and reliably aggregate data across studies, improving diagnostic accuracy, fairness in assessment across different populations, and the comparability of research findings. For example, the SAQOL-39 demonstrated strong cross-cultural validity, with alpha coefficients above 0.9 in several studies, ensuring its applicability across diverse populations. Similarly, the LAST exhibited high inter-rater reliability (ICC > 0.8) and robust construct validity when correlated with gold-standard assessments like the BDAE. At the same time, the assessment of content and criterion validity is also crucial, especially in PWAs, as it ascertains that the instrument accurately measures all relevant aspects of the language disorder (e.g., comprehension, expression, reading, writing, etc.) and aligns with established benchmarks. In almost all the founded studies, content and structure validity are carried out by experts in the field (e.g., healthcare professionals such as speech therapists) and, in some cases, by family members/caregivers of the PWA.

Responsiveness (item 10) has been shown to be the least evaluated psychometric property; the only two studies in which its measurement is present is the validation of the Oxford Cognitive Screen (OCS) by Valera-Gran et al. (2018) [211] and the validation of the Stroke and Aphasia Quality of Life Scale–39 (SAQOL-39) by Sommer et al. (2024) [300]. Measuring responsiveness is a complex challenge, as it requires conducting costly and resource-intensive longitudinal studies; moreover, the lack of standardized methods for assessing responsiveness makes its evaluation difficult and less commonly prioritized compared to validity and reliability [90]. Interestingly, some tools, such as the Token Test, have demonstrated both strong criterion validity and responsiveness, making them valuable for tracking longitudinal changes in patients with aphasia. These findings suggest that tools with a well-established responsiveness can offer clinicians a more dynamic perspective on patient progress over time. Additionally, insufficient follow-up data to assess changes over time and the clinical variability of aphasia conditions, where changes depend on multiple factors, further complicate responsiveness testing (this is especially common in studies that focus on initial validation rather than long-term efficacy studies). As a result, responsiveness is often underreported, despite its importance in demonstrating the clinical utility of assessment tools for tracking patient progress.

From this systematic review, 65 studies (36% of the total) were published before 2010, the year of publication of the COSMIN checklist, which provides specific recommendations on the terminology, taxonomy, and methodology in studies dealing with PROMs and their measurement properties. As reported in Table 2, 15 studies date back to the late 1900s, often making it difficult to access the full text of these articles. Moreover, the reporting of psychometric properties is often inadequate compared to current scientific standards. This lack of detail makes it difficult to properly assess the tools’ validity and reliability, which, in turn, hinders their effective comparison and use in clinical practice and research.

This review reveals a wide array of tools available for clinicians and researchers working with PWA. Among the identified tools, the LAST, SAQOL-39, OCS, Functional Outcome Questionnaire for Aphasia (FOQ-A), and Token Test stand out for their robust psychometric properties, making them reliable options in both clinical and research settings. The comprehensive evaluation of these tools supports their application in assessing various dimensions of aphasia, ranging from language deficits to quality of life and caregiver burden.

In conclusion, our systematic review of aphasia assessment scales suggests that, rather than focusing on the development of entirely new scales, it would be more advantageous to prioritize the refinement and deeper understanding of existing, widely used assessment tools. This process involves further exploration of their psychometric properties to enhance their reliability and validity; additionally, it is important to encourage studies that facilitate the comparison of existing outcomes related to aphasia treatment and rehabilitation. Adopting such approaches could support the execution of meta-analyses that integrate findings from various studies, offering a more coherent and robust view of the effectiveness of rehabilitative interventions for aphasia. This integrated approach may ultimately contribute to greater standardization in assessments and provide better guidance for clinical practice and future research.

### Limitations of the Study

This review has several limitations that should be considered. First of all, it is important to note that excluded tools were not necessarily considered invalid but, rather, did not meet the predefined criteria for providing clear information on the analysis and reporting of psychometric properties. Moreover, our review focused solely on published studies, which means that unpublished research, studies under review, or those accepted for publication only recently were not included. Additionally, some studies may not have been identified with the search strategy used.

This systematic review revealed significant inconsistencies among the currently available aphasia assessment tools. The limitations of this study stem from the difficulty in standardizing the way studies on aphasia are reported, which makes it challenging to compare their findings and to establish consistent norms and standards for assessing language impairment in PWA. We believe that the further exploration, analysis, and refinement of existing assessment instruments are needed rather than the development of new aphasia assessment tools.

## 5. Conclusions

To our knowledge, this is the first systematic review that examines the assessment tools used to assess language impairment in adults with aphasia, considering key parameters such as validity, reliability, responsiveness, and the availability of these tools in various languages, according to the COSMIN checklist. For the advancement of clinical practice and research, it is crucial that we establish practical and widely accepted assessment measures for aphasia. Such consensus would enable the comparison of results across studies and facilitate high-quality meta-analyses of randomized controlled trials involving PWA. Currently, there is no universally accepted assessment tool for aphasia that allows for consistent comparisons of study outcomes. This review highlights the urgent need for agreement among researchers regarding which existing tools should be further studied, adapted to different cultural contexts, and standardized to establish universal norms for assessing aphasia in its various domains.

## Figures and Tables

**Figure 1 brainsci-15-00271-f001:**
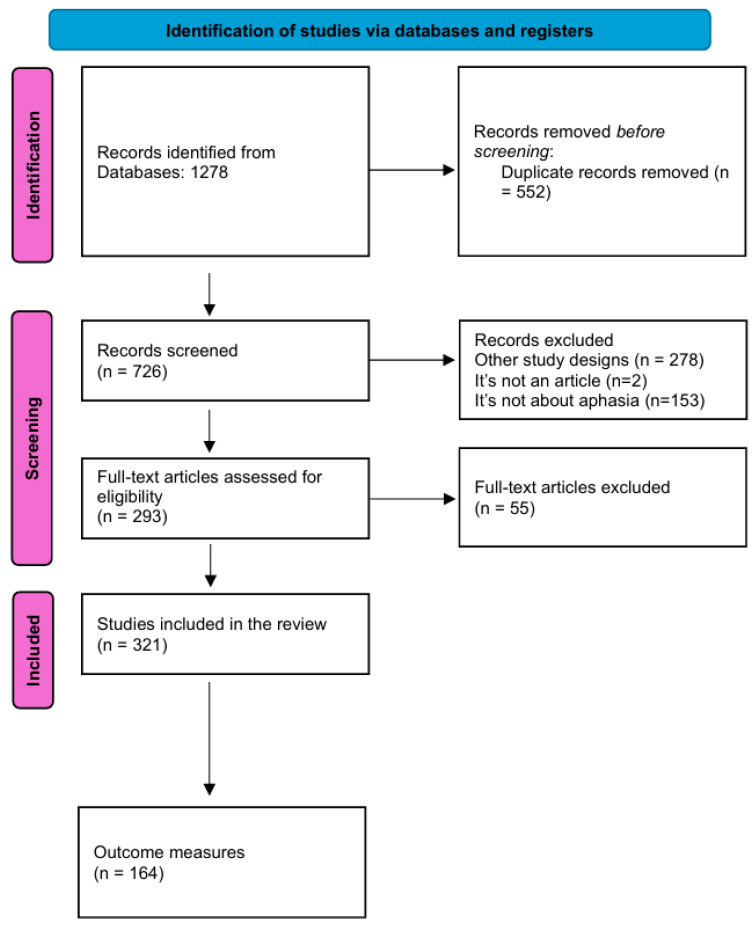
Flowchart of search and screening process.

**Table 1 brainsci-15-00271-t001:** Categorization of identified tools.

Categorization	Identified Tools
Diagnostic tools	Aachen Aphasia Test (AAT)
	An Object and Action Naming Battery (An O&A Battery)
	Auditory Comprehension Test for Sentences (ACTS)
	Batterie d’évaluation de la compréhension syntaxique (BEPS)
	Bilingual Aphasia Test (BAT)
	Boston Diagnostic Aphasia Examination (BDAE)
	Boston Diagnostic Aphasia Examination—Short Form (BDAE-SF)
	Boston Naming Test (BNT)
	Categorical Naming Test (CNT)
	Core Assessment of Language Processing (CALAP)
	Comprehensive Aphasia Test (CAT)
	Comprehensive Assessment of Reading in Aphasia (CARA)
	Computerized Language Analysis (CLAN)
	Confrontation Naming Test (CNT)
	Controlled Oral Word Association Test (COWAT)
	Cracow Neurolinguistic Battery of Aphasia Examination (CN-BAE)
	Diagnostic Aphasia Battery (DAB)
	Diagnostic Instrument for Mild Aphasia (DIMA)
	Ege Aphasia Test (EAT)
	Frontal Behavioral Inventory (FBI)
	Hungarian Aphasia Screening Test (HAST)
	Language Assessment Test for Aphasia (LATA)
	Language Screening Test (LAST)
	Lothian Assessment for screening cognition in Aphasia (LASCA)
	Mental Deterioration Battery (MDB)
	Mini Mental State Examination (MMSE)
	Mississippi Aphasia Screening Test (MAST)
	Montreal Cognitive Assessment (MoCA)
	Montreal-Toulouse Language
	Assessment Battery (MTL-BR)
	Brief Montreal-Toulouse Language
	Assessment Battery (MTL-BR)
	Multiple-Choice Test of Auditory Comprehension for Aphasia (MCTAC)
	Multilingual Aphasia Examination (MAE)
	Multilingual Aphasia Examination—Visual Naming Test (MAE-VNT)
	Naming Assessment in Multicultural Europe (NAME)
	Exploration of Natural Metalinguistic Skills in Aphasia (MetAphAs)
	Neuro-Cognitive Assessment Battery for Stroke Patients (NCABS)
	Non-language based Cognitive Assessment (NLCA)
	Northwestern Anagram Test (NAT)
	Northwestern Assessment of Verbs and Sentences (NAVS)
	Northwestern Assessment of Verbs and Sentences—Verb Naming Test (NAVS-VNT) subtest
	Object and Action Naming Battery: Objects (OANBObj) and Object and Action Naming Battery: Actions (OANBAct)
	Oxford Cognitive Screen (OCS)
	Paced Auditory Serial Addition Test (PASAT)
	Philadelphia Brief Assessment of Cognition (PBAC)
	Philadelphia Naming Test (PNT)
	Picture Naming Test (PNT)
	Preliminary Neuropsychological Battery (PNB)
	Prueba Argentina Psicolingüística de Denominación de Imágenes (PAPDI)
	Psycholinguistic Assessments of Language Processing in Aphasia (PALPA)
	Pyramids and Palm Trees Test (PPT)
	Quick aphasia battery (QAB)
	Reading Comprehension Battery in Aphasia (RCBA)
	Repeatable Battery for the Assessment of Neuropsychological Status (RBANS)
	Russian Aphasia Test (RAT)
	ScreeLing (SL)
	Screening Aphasia Test (SAT)
	Screening Léxico para las Afasias (SLA)
	Screening for Aphasia in NeuroDegeneration (SAND)
	Screening instrument for neuropsychological impairment in stroke (SINS)
	Screening Test for Aphasia and Dysarthria (STAD)
	Sentence Repetition Test (SRT)
	Severity-Calibrated Aphasia Naming Test (SCANT)
	Short Aphasia Test for Gulf Arabic speaking populations (SATG)
	Size/Weight Attribute Test (SWAT)
	Standard Language Test of Aphasia (SLTA)
	Sydney Language Battery (SYDBAT)
	TEst Français de RÉpétition de Phrases (TEFREP)
	Test de dénomination d’actions par vidéos (T-DAV)
	Test de Dénomination de Québec-30 images (TDQ-30)
	Token Test (TT)
	Verb and Sentence Test (VAST)
	Visual-Analogue Test Assessing Anosognosia for Language Impairment (VATA-L)
	Western Aphasia Battery (WAB)
	Western Aphasia Battery—Revised (WAB-R)
	Western Aphasia Battery—Language Quotient (WAB-LQ)
	Western Aphasia Battery—Revised (WAB-R) Picture Description Task
Screening tools	Aachener Aphasie Bedside Test (AABT)
	Acute Aphasia Screening Protocol (AASP)
	Addenbrooke’s Cognitive Examination (ACE)
	Mini—Addenbrooke’s Cognitive Examination (Mini-ACE)
	Aphasia Check List (ACL)
	Aphasia Rapid Test (ART)
	Azeri aphasia screening test
	Bedside Aphasia Battery (BAB)
	Bedside Aphasia Screening Test (BAST)
	Birmingham Cognitive Screen (BCoS)
	Brief Aphasia Evaluation (BAE)
	Brief Evaluation of Receptive Aphasia (BERA)
	Brief test of Cognitive-Communication Disorders (BCCD)
	Cognitive assessment scale for stroke patients (CASP)
	Detection Test for Language impairments in Adults and the Aged (DTLA)
	Frenchay Aphasia Screening Test (FAST)
Outcome measures	Abbey Pain Scale (APS)
	American Speech-Language and Hearing Association Functional Assessment of Communication Skills for Adults (ASHA-FACS)
	Amsterdam-Nijmegen Test for Everyday Language (ANELT)
	Aphasia and stroke therapeutic alliance measure (A-STAM)
	Aphasia Communication Outcome Measure (ACOM)
	Aphasia Impact Questionnaire 21 (AIQ)
	Aphasic Depression Rating Scale (ADRS)
	Apraxia of Speech Rating Scale (ASRS)
	Assessment of Living with Aphasia (ALA)
	Assessment of Communicative Effectiveness in Severe Aphasia (ACESA)
	Auditory-Perceptual Rating of Connected Speech in Aphasia (APROCSA)
	Behavioural Outcomes of Anxiety questionnaire (BOA)
	Basic Outcome Measure Protocol for Aphasia (BOMPA)
	Communication Confidence Rating Scale for Aphasia (CCRSA)
	Communication Outcome after Stroke (COAST)
	Carer Communication Outcome after Stroke (Carer COAST)
	Communicative Activities Checklist (COMACT)
	Communicative Access Measures for Stroke (CAMS)
	Communicative Activity Log (CAL)
	Communicative Competence Scale (CCS)
	Communicative Effectiveness Index (CETI)
	Communicative Participation Item Bank (CPIB)
	Community Integration Questionnaire (CIQ)
	Conversation and Communication Questionnaire for People with Aphasia (CCQA)
	Cuestionario para la Evaluación Enfermera de las Capacidades Comunicativas en la Afasia (CEECA)
	Dynamic Visual Analogue Mood Scales (D-VAMS)
	Frontotemporal Dementia Rating Scale (FTD-FRS)
	Functional Communication Scale (FCS)
	Functional Numeracy Assessment (FNA)
	Functional Outcome Questionnaire for Aphasia (FOQ-A)
	Health Professionals and Aphasia Questionnaire (HPAQ)
	La Trobe Communication Questionnaire (LCQ)
	Lille’s Apathy Rating Scale (LARS)
	Language-Specific Attention Treatment (L-SAT)
	Linguistic Communication Measure (LCM)
	Main Concept Analysis (MCA)
	Measure of Stroke Environment (MOSE)
	Measure of Skills in Supported Conversation & Measure of Participation in Conversation (MSC-MPC scales)
	Montreal Evaluation of Communication (MEC)
	Montreal Evaluation of Communication Brief Battery (MEC-B)
	Naming and Oral Reading for Language in Aphasia 6-Point scale (NORLA-6)
	Neuropsychiatric Inventory (NPI)
	Northwestern Narrative Language Analysis (NNLA)
	Pain Assessment Checklist for Seniors with Limited Ability to Communicate (PACSLAC)
	Perceived Stress Scale (PSS)
	Progressive Aphasia Language Scale (PALS)
	Progressive Aphasia Severity Scale (PASS)
	Quality of life questionnaire for aphasics (QLQA)
	Scandinavian Stroke Scale (SSS)
	Scenario Test (ST)
	Shewan Spontaneous Language Analysis (SSLA)
	Signs of Depression Scale (SODS)
	Sickness impact profile (SIP-65)
	Simple Aphasia Stress Scale (SASS)
	Social Activities Checklist (SOCACT)
	Story Retell Procedure (SRP)
	Stroke and Aphasia Quality of Life Scale–39 (SAQOL-39)
	Stroke Aphasic Depression Questionnaire (SADQ)
	Stroke Communication Scale (ICF-SCS)
	Stroke Social Network Scale (SSNS)
	Stroke Specific Quality of Life Scale (SS-QOL)
	Visual Analog Mood Scales (VAMS)
	Visual Analog Mood Scales—Revised (VAMS-R)
	Visual Analogue Self-Esteem Scale (VASES)
	World Retrieval in Aphasic Discourse (WRAD)
Unspecified	Aachener Sprachanalyse (ASPA)
	AphasiaBank Stimuli
	Augmentative and Alternative Communication Assessment Questionnaire (AAC)
	City Gesture Checklist (CGC)
	Closed Answers, Pro-speak question, Simple orders, Common object denomination, Audio repetition, Reading, Evoke (CA-PS CARE)
	Core Lexicon and Microlinguistic Measures
	Decision-Making Capacity Assessments (DMCA)

**Table 3 brainsci-15-00271-t003:** Quality scores of the included studies according to the COSMIN checklist.

Assessment Tool	Author (Year)	Item 1	Item 2	Item 3	Item 4	Item 5	Item 6	Item 7	Item 8	Item 9	Item 10
AAT	Huber W. (1984) [30]	+	n.a.	n.a.	n.a.	n.a.	n.a.	n.a.	n.a.	n.a.	n.a.
	Pracharitpukdee N. (2000) [31]	-	-	-	-	-	+	-	-	-	-
	Miller N. (2000) [32]	-	+	-	-	+	-	-	+	-	-
	Lauterbach M. (2008) [33]	+	+	+	+	+	-	+	+	+	-
	Luzzatti C. (2023) [34]	+	-	+	-	+	-	-	+	-	-
AABT	Biniek R. (1992) [35]	+	n.a.	n.a.	n.a.	n.a.	n.a.	n.a.	n.a.	n.a.	n.a.
	Muò (2021) [36]	+	+	+	+	+	-	-	+	-	-
ASPA	Barthel G. (2006) [37]	+	+	+	+	+	-	-	+	-	-
APS	Abbey J. (2004) [38]	+	+	-	-	-	+	-	-	-	-
	Storti M. (2009) [39]	-	+	+	+	-	+	+	-	-	-
	Takai Y. (2010) [40]	-	+	+	+	-	+	+	-	-	-
	Neville C. (2013) [41]	-	-	+	+	-	+	-	-	-	-
	Gregersen M. (2016) [42]	+	+	+	-	+	+	-	+	-	-
AASP	Crary M.A. (1989) [43]	+	+	+	-	-	+	+	-	-	-
ACE	Hodges J.R. (2017) [44]	n.a.	n.a.	n.a.	n.a.	n.a.	n.a.	n.a.	n.a.	n.a.	n.a.
	Gaber T.A. (2011) [45]	-	+	+	+	+	-	+	+	+	-
	Elamin M. (2016) [46]	-	+	-	-	-	+	+	+	+	-
Mini-ACE	Hsieh S. (2015) [47]	+	-	-	-	-	+	-	-	-	-
ASHA-FACS	de Carvalho I.A.M. (2008) [48]	-	n.a.	n.a.	+	+	+	+	-	n.a.	-
	Muò R. (2015) [49]	-	+	-	-	+	-	+	+	-	-
ANELT	Blomert L. (1994) [50]	+	+	+	-	-	+	+	-	-	-
	Ruiter M.B. (2011) [51]	-	-	+	-	-	-	-	-	-	-
	Ruiter M.B. (2022) [52]	-	-	+	+	+	+	-	-	-	-
	Wong W.W.S. (2024) [53]	-	+	+	+	-	+	+	-	-	-
An O&A Battery	Edmonds (2012) [54]	-	+	+	+	-	+	-	-	-	-
	Spezzano L.C. (2013) [55]	-	-	-	-	-	-	-	-	-	-
A-STAM	Lawton M. (2019) [56]	+	+	-	-	+	-	+	+	-	-
AphasiaBank Stimuli	Boyle M. (2015) [57]	-	-	-	-	-	+	-	+	-	-
ACL	Kalbe E. (2005) [58]	-	+	+	+	+	+	-	+	+	-
	Zadeh A.M. (2021) [59]	+	+	-	-	+	-	+	+	+	-
ACOM	Hula W.D. (2015) [60]	+	+	-	-	-	+	+	+	+	-
AIQ	Swinburn K. (2019) [61]	-	+	-	-	+	-	-	+	+	-
	Yaşar E (2022) [62]	-	+	-	+	+	-	+	+	-	-
ART	Azuar C. (2013) [63]	-	+	-	+	+	-	-	+	-	-
	Panebianco M. (2019) [64]	-	+	-	+	+	-	+	+	-	-
	Buivolova O. (2021) [65]	-	+	-	+	+	-	-	+	+	-
	Kavakci M. (2022) [66]	+	+	+	+	+	+	+	+	+	-
ADRS	Benaim C. (2004) [67]	-	+	-	-	-	-	+	+	-	-
ASRS	Strand E.A. (2014) [68]	+	-	-	-	-	+	-	-	-	-
	Wambaugh J.L. (2019) [69]	-	-	-	+	-	+	+	-	-	-
	Hybbinette H. (2021) [70]	-	-	-	-	-	+	-	-	-	-
	Duffy J.R. (2023) [71]	-	-	+	-	-	+	-	+	-	-
	Santos D.H.N.D. (2023) [72]	-	+	-	-	+	+	-	-	-	-
ALA	Simmons-Mackie N. (2014) [73]	-	+	-	+	+	-	-	+	+	-
	Guo Y.E. (2017) [74]	-	+	+	+	-	+	-	-	+	-
ACESA	Cunningham R. (1995) [75]	-	+	-	-	+	-	-	+	-	-
ACTS	Klor B.M. (1980) [76]	+	+	+	+	+	+	-	+	+	-
	Flanagan J.L. (1997) [77]	-	-	-	-	-	+	+	+	-	-
APROCSA	Casilio M. (2019) [78]	+	+	+	+	+	-	+	+	+	-
AAC	Petrosyan T.R. (2022) [79]	-	+	-	-	-	-	+	+	-	-
Azeri aphasia screening test	Salehi S. (2016) [80]	-	+	-	+	+	-	+	+	-	-
BEPS	Bourgeois M.E. (2019) [81]	+	-	+	+	-	-	-	-	-	-
	Coulombe V. (2021) [82]	+	+	-	-	-	-	-	+	+	-
BAB	Sivagnanapandian D. (2022) [83]	-	+	-	+	+	-	-	+	+	-
BAST	Cruz A.L. (2014) [84]	-	+	-	-	-	-	-	+	+	-
BOA	Eccles A. (2017) [85]	+	+	+	+	+	-	+	+	+	-
BAT	Amberber A.M. (2011) [86]	+	+	+	-	+	+	-	+	+	-
	Gomez-Ruiz I (2011) [87]	+	+	+	+	+	-	-	+	-	-
	Peristeri E. (2011) [88]	-	-	-	-	-	+	-	-	-	-
	Amberber A.M. (2012) [89]	+	+	-	+	+	-	+	+	-	-
	Krishnan G. (2017) [90]	-	+	-	-	-	-	-	+	-	-
BCoS	Pan X. (2015) [91]	+	+	-	-	+	+	+	+	-	-
	Kong A.P.H. (2018) [92]	+	+	-	+	-	-	+	+	+	-
	Kuzmina E. (2018) [93]	-	+	+	+	+	+	+	+	+	-
BOMPA	Kagan A. (2020) [94]	+	-	-	-	-	+	-	-	-	-
BDAE	Pineda D.A. (2000) [95]	-	+	+	+	+	-	-	+	+	-
	Fong M.W.E. (2019) [96]	+	+	+	+	+	-	+	+	+	-
BDAE-SF	Flanagan J.L. (1997) [77]	-	-	-	-	-	+	+	+	-	-
	Tsapkini K. (2009) [97]	-	-	-	-	-	-	-	-	-	-
	Del Toro C.M. (2011)	+	+	-	+	+	-	+	+	-	-
BNT	Del Toro C.M. (2011) [96]	+	+	-	+	+	-	+	+	-	-
	Aniwattanapong D. (2019) [97]	-	-	-	+	-	+	-	-	-	-
	Sachs A. (2020) [98]	-	+	-	-	+	-	-	+	+	-
BAE	Vigliecca N.S. (2011) [99]	-	+	+	+	+	+	-	+	+	-
	Vigliecca. N.S. (2019) [100]	-	+	-	-	+	+	-	+	-	-
BERA	Aubinet C. (2021) [101]	-	+	-	+	-	+	-	+	-	-
BCCD	Lee M.S. (2020) [27]	+	+	-	+	+	-	+	+	-	-
CNT	Hwang Y.M. (2021) [102]	+	+	-	+	+	-	-	+	+	-
CGC	Caute A. (2021) [103]	+	+	-	+	+	-	-	+	+	-
CA-PS CARE	Ferri L. (2021) [104]	-	+	+	+	+	-	+	+	+	-
CASP	Park K.H. (2017) [105]	+	+	+	+	-	-	+	+	-	-
	Benaim C. (2022) [106]	-	+	+	+	+	-	+	+	-	-
CALAP	Jacquemot C. (2019) [107]	+	+	+	+	+	-	+	+	+	-
Core Lexicon and Microlinguistic Measures	Kim H. (2019) [108]	+	+	+	-	-	+	+	+	+	-
CCRSA	Cherney L.R. (2011) [109]	+	+	-	-	-	-	-	+	-	-
COAST	Long A.F. (2009) [110]	+	+	+	+	+	+	+	+	-	-
	Bambini V. (2017) [111]	+	+	-	+	+	-	+	+	-	-
Carer COAST	Long A.F. (2009) [110]	+	+	+	+	-	+	-	-	+	-
COMACT	Aujla S. (2015) [112]	-	+	+	+	-	+	-	-	-	-
CAMS	Kagan A. (2017) [113]	+	+	+	+	-	+	-	-	-	-
CAL	Kim D.Y. (2019) [114]	-	+	+	+	+	+	+	+	+	-
	Habili M. (2022) [115]	+	+	+	+	+	-	-	+	+	-
CCS	Brock K.L. (2019) [116]	-	+	-	+	+	+	+	+	+	-
CETI	Lomas J. (1989) [117]	+	n.a.	n.a.	+	-	+	+	n.a.	n.a.	n.a.
	Pedersen P.M. (2001) [118]	-	+	-	-	+	+	+	+	-	-
	Moretta P. (2021) [119]	-	+	+	-	+	+	+	+	+	-
	Charalambous M. (2024) [120]	-	+	+	+	+	+	-	+	+	-
CPIB	Baylor C. (2017) [121]	-	-	-	-	-	-	-	-	-	-
CIQ	Dalemans R.J. (2010) [122]	-	+	+	+	+	-	+	+	+	-
CAT	Abou El-Ella M.Y. (2013) [123]	+	+	-	-	-	-	+	+	-	-
	Maviş İ. (2022) [124]	-	+	+	-	-	-	+	+	+	-
	Zakariás L. (2022) [125]	-	+	+	+	+	+	-	+	+	-
	Kong A.P.H. (2022) [126]	-	n.a.	n.a.	n.a.	+	n.a.	n.a.	n.a.	n.a	n.a
	Jensen B.U. (2024) [127]	-	+	-	-	-	+	+	+	+	-
CARA	Thumbeck S.M. (2023) [128]	-	+	+	+	+	-	-	+	+	-
CLAN	Hsu C.J. (2018) [129]	-	-	-	-	-	+	-	-	-	-
CNT	Vigliecca N.S. (2019) [130]	-	+	-	-	+	-	+	+	+	-
COWAT	Ross T.P. (2003) [131]	-	-	-	-	-	+	-	-	-	-
CCQA	Horton S. (2020) [132]	-	+	-	+	+	-	-	+	-	-
CN-BAE	Pachalska M. (1995) [133]	-	+	-	+	+	-	-	+	-	-
CEECA	Martín-Dorta W.J. (2023) [134]	-	+	+	+	+	+	+	+	+	-
DMCA	Carr F.M. (2023) [135]	+	+	-	-	+	-	+	+	-	-
DTLA	Macoir J. (2017) [136]	-	+	-	-	+	-	+	+	+	-
	Karalı F.S. (2024) [137]	-	+	-	+	-	+	-	-	-	-
DAB	Al-Thalaya Z. (2018) [138]	+	+	+	-	+	+	-	+	+	-
DIMA	Clément A. (2022) [139]	+	+	+	-	+	+	+	+	+	-
D-VAMS	Barrows P.D. (2018) [140]	+	-	+	+	-	+	+	-	-	-
EAT	Atamaz F. (2007) [141]	+	+	+	+	+	+	-	+	+	-
	Calis F.A. (2013) [142]	-	+	+	+	+	+	+	+	-	-
FAST	Enderby P. (1996) [143]	-	-	+	+	+	+	+	+	+	-
	Paplikar A. (2020) [144]	-	-	-	+	-	+	-	-	-	-
FBI	Kertesz A. (2000) [145]	-	+	+	+	+	+	-	+	-	-
FTD-FRS	Lima-Silva T.B. (2013) [146]	-	-	-	-	+	-	-	-	-	-
	Turró-Garriga O. (2017) [147]	-	+	+	+	-	-	+	-	-	-
	Lima-Silva T.B. (2018) [148]	-	+	+	+	-	-	+	+	-	-
FCS	Drummond S.S. (2004) [149]	+	+	-	-	-	-	+	+	-	-
FNA	Ichikowitz K. (2022) [150]	+	-	-	+	-	+	+	-	-	-
FOQ-A	Glueckauf R.L. (2003) [151]	+	+	+	-	+	+	-	+	+	-
	Ketterson T.U. (2008) [152]	-	+	-	+	+	+	+	+	-	-
	Mitasova A. (2015) [153]	n.a.	n.a.	n.a.	n.a.	n.a.	n.a.	n.a.	n.a.	n.a.	n.a.
	Spaccavento S. (2018) [154]	-	+	-	+	+	-	+	+	+	-
HPAQ	Jensen L.R. (2022) [155]	+	+	-	-	+	-	+	+	+	-
HAST	Zakariás L. (2023) [156]	+	+	+	+	-	+	-	-	-	-
LCQ	Douglas J.M. (2007) [157]	-	+	+	+	+	-	+	+	+	-
	Struchen M.A. (2008) [158]	-	-	+	+	-	+	-	-	-	-
LATA	Toǧram B. (2012) [159]	-	+	+	-	+	+	+	+	-	-
LARS	Fernández-Matarrubia M. (2015) [160]	+	+	-	-	+	-	+	+	+	-
LAST	Flamand-Roze C. (2011) [161]	-	+	-	+	+	-	+	+	-	-
	Burgeois-Marcotte J. (2015) [162]	-	+	+	+	+	-	-	+	-	-
	Flowers H.L. (2015) [163]	-	+	-	-	-	+	-	-	-	-
	Koening-Bruhin M. (2016) [164]	-	+	+	+	+	+	+	+	-	-
	Yang H. (2018) [165]	-	-	+	+	+	+	-	-	-	-
	Sun M. (2020) [166]	n.a.	n.a.	n.a.	n.a.	n.a.	n.a.	n.a.	n.a.	n.a.	n.a.
	Ramos R.D.L. (2023) [167]	-	+	+	+	+	+	-	-	-	-
L-SAT	Peach R.K. (2018) [168]	-	+	+	+	+	+	+	+	-	-
LCM	Kong A.P.H (2009) [26]	-	+	+	+	+	+	+	+	+	-
	Kong A.P.H (2009) [26]	-	+	-	+	+	-	+	+	+	-
LASCA	Faiz A. (2016) [169]	-	+	+	+	+	+	-	+	-	-
MCA	Kong A.P.H. (2015) [170]	-	+	+	-	+	-	-	+	+	-
	Yazu H. (2022) [171]	-	+	+	+	+	+	+	+	-	-
MOSE	Babulal G.M. (2016) [172]	-	+	+	-	+	+	+	+	+	-
	Wang W. (2022) [173]	-	+	+	+	+	+	-	-	+	-
MSC-MPC scales	Kagan A. (2004) [94]	+	-	-	-	-	-	-	-	-	-
	Muò R. (2019) [174]	-	-	-	-	+	+	+	-	-	-
MDB	Carlesimo G.A. (1996) [175]	-	+	-	-	+	+	+	+	-	-
MMSE	Vigliecca N.S. (2012) [176]	-	+	+	+	+	+	-	+	+	-
MAST	Nakase-Thompson R. (2005) [177]	+	-	-	-	-	+	+	+	-	-
	Kostálová M. (2008) [178]	-	+	+	+	+	+	+	+	-	-
	Romero M. (2012) [179]	-	+	+	+	-	-	+	+	-	-
	Khatoonabadi A.R. (2015) [180]	-	+	-	+	+	-	+	+	-	-
	Nursi A. (2019) [181]	-	-	+	-	+	-	+	+	+	-
MoCA	Lim P.A. (2016) [182]	-	+	+	+	+	+	-	+	-	-
MEC	Le Dorze G. (2000) ) [183]	+	+	+	-	-	+	+	-	-	-
MEC-B	Casarin F.S. (2019) [184]	+	-	+	+	-	+	-	-	-	-
MTL-B	Pagliarin K.C. (2014) [185]	+	-	-	-	-	-	+	-	-	-
	Pagliarin K.C. (2015) [186]	-	-	-	-	-	+	+	-	-	-
MTL-BR	Altmann R.F. (2020) [187]	+	+	-	-	-	-	-	-	-	-
MCTAC	Hallowell B. (2009) [188]	-	+	+	+	+	-	-	+	+	-
MAE	Rey G.J. (2001) [189]	-	+	+	+	+	+	+	+	+	-
MAE-VNT	Axelrod B.N. (1994) [190]	-	-	-	-	-	-	-	+	-	-
NORLA-6	Pitts L.L. (2018) [191]	-	+	-	+	+	-	-	+	+	-
NAME	Franzen S. (2023) [192]	+	+	+	+	+	-	-	+	-	-
MetAphAs	Mac-Kay A.P.M.G. (2020) [193]	+	+	-	+	+	+	-	+	-	-
NCABS	Mahmood S.N. (2018) [194]	-	+	+	+	+	-	-	+	-	-
NPI	Yiannopoulou K.G. (2019) [195]	-	+	+	+	+	-	+	+	-	-
NLCA	Wu J.B. (2017) [196]	-	+	-	+	+	-	+	+	+	-
NAT	Weintraub S. (2009) [197]	+	+	+	+	+	-	+	+	+	-
NAVS	Thompson C. (2012) [198]	-	-	-	-	-	+	-	-	-	-
	Cho-Reyes S. (2012) [199]	-	-	-	-	-	-	-	-	-	-
NAVS-VNT subtest	Wang H. (2016) [200]	-	-	+	+	-	+	-	+	-	-
NNLA	Fergadiotis G. (2023) [201]	-	-	-	-	-	+	-	-	-	-
OANBObj/ OANBAct	Hsu C.J. (2017) [129]	n.a.	n.a.	n.a.	n.a.	n.a.	n.a.	n.a.	n.a.	n.a.	n.a.
OCS	Peach R.K. (2018) [168]	-	+	+	+	+	+	+	+	+	-
	Demeyere N. (2015) [202]	-	+	+	+	+	+	-	+	+	-
	Kong A.P. (2016) [203]	-	-	-	-	-	-	-	-	-	-
	Ramos C.C.F. (2018) [204]	-	-	+	+	-	+	+	-	-	+
	Valera-Gran D. (2018) [205]	-	+	+	+	-	+	+	+	+	-
	Hong W. (2018) [206]	-	-	+	+	-	+	-	-	+	-
	Shendyapina M. (2018) [207]	-	+	+	+	+	+	+	+	+	-
	Huygelier H. (2022) [208]	-	-	+	-	-	-	-	-	-	-
	Webb S.S. (2022) [209]	-	-	-	-	-	+	+	-	-	-
	Bormann T. (2023) [210]	-	-	-	-	-	+	-	-	-	-
	Murphy D. (2023) [211]	-	-	-	-	+	-	-	-	-	-
PASAT	Cho E. (2024) [212]	-	+	+	+	+	-	+	+	+	-
PACSLAC	Nikravesh M. (2017) [213]	+	+	+	+	-	+	+	-	-	-
	Fuchs-Lacelle S.M.A. (2004) [214]	-	-	-	-	-	+	-	-	-	-
	Aubin M. (2007) [215]	-	+	+	+	-	+	-	-	-	-
	Cheung G. (2008) [216]	-	-	+	+	-	+	-	-	-	-
	Kim E.K. (2014) [217]	-	-	-	+	-	+	-	+	+	-
	Chan S. (2014) [218]	-	+	+	+	+	+	+	-	-	-
	Thé K.B. (2016) [29]	-	-	+	+	+	+	-	-	-	-
	Büyükturan Ö. (2018) [219]	-	+	+	+	+	+	+	-	-	-
	Haghi M. (2020) [220]	-	+	+	+	+	-	-	+	+	-
PSS	De Vries N.J. (2023) [221]	+	+	-	+	+	-	+	+	+	-
PBAC	Pompon R.H. (2018) [222]	+	-	-	+	-	+	+	-	-	-
PNT	Libon D.J. (2011) [223]	-	+	+	+	+	+	-	+	-	-
	Walker G.M. (2012) [224]	-	+	+	-	+	+	+	+	+	-
PhNT	Swiderski A.M. (2023) [225]	-	+	-	+	+	-	+	+	-	-
	Tahanzadeh B. (2017) [226]	-	+	-	-	+	+	n.a.	n.a.	n.a.	n.a.
	Macoir J. (2021) [227]	-	+	+	+	+	-	+	+	+	-
	Vivas L. (2022) [228]	-	+	+	+	-	+	-	+	+	-
PNB	Khatibani M.N. (2022) [229]	-	+	-	-	+	+	-	+	+	-
PALS	Hobson J.P. (2003) [230]	-	+	-	+	+	+	-	+	+	-
PASS	Jafari S. (2018) [231]	-	-	-	-	-	+	-	-	-	-
	Sapolsky D. (2010) [232]	-	n.a.	n.a.	n.a.	+	n.a.	n.a.	n.a.	n.a.	n.a.
PAPDI	Petrillo S.M. (2022) [233]	-	+	-	-	+	+	+	+	+	-
PALPA	Manoiloff L. (2018) [234]	+	+	+	+	+	+	+	+	+	-
PPT	Pinto-Grau M. (2021) [235]	-	+	-	+	+	+	+	+	+	-
QLQA	Klein L.A. (2009) [28]	-	+	+	+	+	+	-	+	-	-
QAB	Spaccavento S. (2014) [5]	-	+	+	+	+	+	-	+	-	-
	Wilson S.M. (2018) [236]	-	+	+	+	+	+	+	-	-	-
	Zhu D. (2023) [237]	-	-	-	+	-	+	+	-	-	-
RCBA	Parlak M.M. (2024) [238]	-	+	-	-	+	-	+	+	-	-
	Van Demark A.A. (1982) [239]	-	-	-	-	-	+	+	+	-	-
RBANS	Flanagan J.L. (1997) [77]	-	+	+	+	+	-	-	+	+	-
	Wilde M.C. (2006) [240]	-	+	-	-	-	-	-	+	+	-
RAT	Merz Z.C. (2018) [241]	-	+	-	-	-	+	+	+	+	-
SSS	Ivanova M.V. (2021) [242]	-	+	-	-	-	-	+	+	+	-
ST	Thommessen B. (2002) [243]	-	+	-	+	+	-	-	+	-	-
	Hilari K (2018) [244]	-	+	+	+	+	+	+	+	-	-
	van der Meulen I. (2010) [245]	-	+	-	+	+	-	+	+	-	-
	Charalambous M. (2022) [246]	-	+	-	+	-	+	-	-	+	-
SL	Kodani Y. (2024) [247]	-	+	-	+	+	-	-	+	+	-
SAT	El Hachioui H. (2012) [248]	-	+	+	+	+	-	+	+	-	-
	Abou-Elsaadd T. (2018) [249]	-	-	-	-	-	-	-	-	-	-
SLA	Banco E. (2023) [250]	+	-	+	-	-	+	+	-	-	-
SAND	Peña-Chávez R. (2014) [251]	+	-	-	-	-	-	-	-	-	-
	Catricalà E. (2018) [252]	-	+	+	+	+	-	+	+	+	-
	Battista P. (2018) [253]	+	+	+	+	-	-	+	+	+	-
SINS	Picillo M. (2019) [254]	+	+	-	-	-	-	-	+	-	-
STAD	Sødring K.M. (1998) [255]	+	-	+	-	-	-	-	-	+	-
SRT	Araki K. (2022) [256]	+	n.a.	n.a.	n.a.	n.a.	n.a.	n.a.	n.a.	n.a.	n.a.
	Spreen O. (1963) [257]	+	+	-	-	+	-	+	+	+	-
SCANT	Meyers J.E. (2000) [258]	-	+	+	+	+	+	+	+	+	-
SSLA	Walker G.M. (2022) [259]	+	+	+	-	-	+	-	+	-	-
SATG	Shewan C.M. (1988) [260]	-	+	+	+	+	+	+	+	-	-
SODS	Altaib M.K. (2021) [261]	+	-	-	+	-	+	+	-	-	-
	Bennett H.E. (2006) [262]	-	-	-	+	-	+	-	-	-	-
	van Dijk M.J. (2017) [263]	-	-	+	+	-	+	-	-	+	-
SIP-65	van Dijk M.J. (2018) [264]	-	+	+	+	+	+	+	+	+	-
	Bénaim C. (2003) [265]	-	-	-	+	+	-	+	+	+	-
SASS	van Dijk M.J. (2018) [264]	+	-	-	-	-	+	+	-	-	-
SWAT	Laures-Gore J. (2019) [266]	+	-	-	-	-	+	+	-	-	-
COMACT	Yoo Y. (2016) [267]	-	+	+	+	-	+	-	-	-	-
SLTA	Aujla S. (2015) [112]	-	n.a.	n.a.	n.a.	n.a.	n.a.	+	n.a.	n.a.	n.a.
SRP	Tsutsumiuchi K. (2020) [268]	+	-	-	-	-	+	-	+	-	-
	Doyle P.J. (2001) [269]	-	-	-	-	-	+	-	+	+	-
	McNeil M.R. (2002) [270]	-	-	-	-	-	+	+	-	-	-
	Hula W.D. (2003) [271]	-	-	-	-	-	+	+	+	+	-
	McNeil M.R. (2002) [270]	-	n.a.	n.a.	n.a.	+	n.a.	n.a.	n.a.	n.a.	n.a.
SAQOL-39	Ortiz K.Z. (2024) [272]	+	+	-	+	+	-	+	+	-	-
	Hilari K. (2003) [273]	-	+	-	+	+	+	+	+	-	-
	Posteraro L. (2004) [274]	-	+	-	+	+	+	-	+	+	-
	Hilari K. (2007) [275]	-	-	-	+	+	+	-	+	-	-
	Hilari K. (2009) [276]	-	+	-	+	+	-	+	+	+	-
	Lata-Caneda M.C. (2009) [277]	-	+	-	+	+	+	+	+	+	-
	Efstratiadou E.A. (2012) [278]	-	+	-	-	+	+	-	+	+	-
	Kiran S. (2013) [279]	-	+	-	+	+	-	+	+	-	-
	Mitra I.H. (2015) [280]	-	+	+	-	+	+	-	+	-	-
	Kamiya A. (2015) [281]	-	+	+	+	+	-	-	+	+	-
	Calis A.F. (2016) [282]	-	+	+	+	+	-	+	+	+	-
	Guo Y.E. (2016) [283]	-	+	+	+	+	+	-	+	+	-
	Noyan-ErbaŞ A. (2016) [284]	-	+	+	+	+	-	+	+	+	-
	Van Ewijk L. (2017) [285]	-	+	+	+	+	-	-	+	+	-
	Qiu W. (2019) [286]	-	+	-	+	+	-	-	+	+	-
	Van Ewijk L. (2019) [287]	-	+	-	+	+	+	+	+	-	-
	Kariyawasam P.N. (2020) [288]	-	+	-	-	+	+	+	+	-	-
	Kristinsson S. (2021) [289]	-	+	+	+	+	+	+	+	-	-
	Azizbeigi-Boukani J. (2021) [290]	-	+	-	-	+	+	-	+	+	-
	Vuković M. (2022) [291]	-	-	-	+	-	+	-	-	+	+
SADQ	Sommer J.B. (2024) [292]	+	-	+	+	-	+	-	-	-	-
	Sutcliffe L.M. (1998) [293]	-	-	-	-	-	-	-	-	-	-
	Sackley C.M. (2006) [294]	-	-	-	-	-	-	-	-	-	-
	Hacker V.L. (2010) [295]	-	-	-	+	+	-	-	-	-	-
ICF-SCS	Cobley C.S. (2012) [296]	+	+	-	-	-	+	-	-	-	-
SSNS	Batista Dallaqua G. (2019) [297]	-	+	+	-	+	-	-	+	+	-
SS-QOL	Northcott S. (2013) [298]	+	+	-	+	+	+	+	+	+	-
SYDBAT	Hilari K. (2001) [299]	+	+	-	-	-	-	+	+	+	-
	Savage S. (2013) [300]	+	+	-	-	+	-	+	+	-	-
TEFREP	Janseen N. (2022) [301]	-	+	-	-	+	-	-	+	+	-
T-DAV	Bourgeois-Marcotte J. (2015) [302]	+	+	-	+	-	+	-	-	-	-
	Spigarelli M. (2022) [303]	n.a.	n.a.	n.a.	n.a.	n.a.	n.a.	n.a.	n.a.	n.a.	n.a.
TDQ-30	Macoir J. (2023) [304]	+	+	-	+	-	+	+	-	-	-
TT	Macoir J. (2019) [227]	-	+	-	+	+	-	-	+	+	-
	Park G.H. (2000) [305]	-	+	+	+	+	-	+	+	+	-
	Hula W. (2006) [306]	-	+	+	+	+	-	-	+	+	-
	Bakhtiar M. (2020) [307]	-	+	+	+	+	+	+	+	-	-
	McNeil M.R. (2015) [308]	-	+	-	+	+	+	-	+	-	-
VAST	Paci M. (2015) [309]	-	+	-	+	+	-	-	+	-	-
VAMS	Bakhtiar M. (2020) [307]	-	-	+	-	-	-	-	-	+	-
	Bastiaanse R. (2003) [310]	-	+	-	+	+	+	+	+	-	-
	Temple R.O. (2004) [311]	+	+	+	+	+	+	+	+	-	-
VAMS-R	Kontou E. (2012) [312]	+	-	-	+	-	-	-	-	-	-
VASES	Barrows P.D. (2018) [313]	+	+	+	+	+	+	+	+	-	-
	Kontou E. (2012) [312]	+	n.a.	n.a.	+	n.a.	+	n.a.	n.a.	n.a.	n.a.
VATA-L	Brumfitt S.M. (1999) [314]	-	+	-	+	+	-	-	+	+	-
WAB	Bennett H.E. (2006) [262]	-	+	+	-	+	+	-	+	+	-
	Cocchini G. (2010) [315]	-	+	+	+	+	+	+	+	+	-
	Shewan C.M. (1980) [316]	-	-	-	-	-	-	+	+	+	-
	Kim H. (2004) [317]	-	+	-	-	-	-	-	+	+	-
	Bakheit A.M.O. (2005) [318]	-	+	+	+	+	-	+	+	+	-
	Hula W. (2010) [319]	-	+	-	+	+	-	+	+	+	-
WAB-R	Neves Mde B. (2014) [320]	-	+	-	+	-	+	+	-	-	-
	Boucher J. (2022) [321]	-	-	-	-	-	+	-	-	-	-
	Nilipour R. (2014) [322]	-	+	+	+	+	+	+	-	-	-
	Dekhtyar M. (2020) [323]	-	-	-	-	-	+	-	-	-	-
WAB-LQ	El Ouardi L. (2023) [324]	+	n.a.	n.a.	n.a.	n.a.	+	n.a.	n.a.	n.a.	n.a.
WAB-R Picture Description Task	Rao L.A. (2022) [325]	-	+	-	-	-	+	-	+	-	-
WRAD	Shewan C.M. (1986) [326]	-	-	-	-	-	+	-	-	-	-

Item 1: PROM development. Item 2: Content validity. Item 3: Structural validity. Item 4: Internal consistency. Item 5: Cross-cultural validity/Measurement invariance. Item 6: Reliability. Item 7: Measurement error. Item 8: Criterion validity. Item 9: Hypotheses testing for construct validity. Item 10: Responsiveness. +: the study meets the required quality standard; -: the study does not meet the required quality standard; n.a.: not available.

## Data Availability

Data supporting this study’s findings are available from the corresponding author upon reasonable request due to privacy and ethical reasons.

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
