# Peer review of "Quality of Assessment Tools for Aphasia: A Systematic Review"

_brainsci, 2025, doi:10.3390/brainsci15030271_

Round 1
Reviewer 1 Report
Comments and Suggestions for Authors
In the paper Measurement properties of outcome measures for aphasia: a systematic review the authors wanted to identify currently available validated measurement tools analyzing their validity and psychometrics properties.
I appreciate the effort the authors have made in compiling the systematic review, but I think the aim of the study is misleading and should be refined – especially part currently available and validated measurements tools. I know there are quite a few other papers that show the psychometric properties of adapted tests, but they are not included in your analysis (for example: Jensen, B. U., Norvik, M. I., & Simonsen, H. G. (2023). Statistics and psychometrics for the CAT-N: Documenting the Comprehensive Aphasia Test for Norwegian. Aphasiology, 38(3), 412–439. https://doi.org/10.1080/02687038.2023.2200132
Therefore, it is not correct to say that you have analyzed currently available, when you have not. It would be more correct to say that you analyzed those that satisfied your inclusion criteria although some of them you were not included because they did not meet them. Here we have two problems:
1) However, this does not mean that the tests you did not consider are not psychometrically valid and reliable and that everything you used as inclusion criteria does not exist in the manuals of these tests. In other words: What psychometric characteristics the authors include in their article is up to them. Therefore, it is not correct to say that your aim is to show which tests are valid, because some published tests are mislabeled in this way. Here I ask a rhetorical question: are these articles really reliable sources of data on the validity and reliability of tests?
2) Moreover, systematic reviews are not merely a listing of articles, but a critical examination of this listing and a critical evaluation and orientation for the future. Accordingly, the introduction lacks the author's critical review of what mandatory psychometric measures each test should take, with a clear explanation of why. The authors cover this important part using COSMIN, which is good, but they do not provide a theoretical background to justify the importance of psychometric measures. The authors treat psychometric measures very poorly. For example, it states: Reliability, the extent to which scores for patients who have not changed are the same for repeated measurement within- rater, between-rater, and over time (lines 159 and 160) - implying that it does not matter what the author wants to measure: Inter-rater reliability or test-retest reliability. It is not the same! Moreover, these two types of reliability are not even expected to have the same reliability coefficients, as they are influenced by different factors. I suggest that the authors already at the end of their introduction give their proposal for the standard of psychometric measures that they think that authors of the tests must provide in such articles to confirm the test validity and to explain each measure.
Specific comments:
Line 57 – something missing in the number – about 1.5 people – I checked the reference in which authors stated that about 15 million people worldwide are affected by aphasia.
Line 172 – indicate the period in which the articles were published – you only indicate this in the discussion (from 1963 to 2024)
Lines 244-246: Therefore, the aim of this study was to provide clinicians and researchers with evidence based recommendations regarding the existing outcome measures for the measurement of aphasia, as well as their psychometric properties – this is a new aim that does not correspond to the aim in lines 72-75; it is not possible to open a new aim in the discussion.
Line 256 – 52,5% of the 181 tools having only a single development or validation article - Is there a rule somewhere about how many articles on the validity of tests authors must publish? Is the manual not a sufficient source for this?
Lines 260-261: many of these instruments do not demonstrate adequate methodological quality, further limiting their reliability and validity in clinical and research settings.- have you provided your own critical and scientifically sound evidence of methodological quality in introduction before drawing this conclusion?
307-309: In conclusion, our systematic review of aphasia assessment scales suggests that rather than continuing to develop new scales, it would be more advantageous to focus on refining and deepening the understanding of existing and widely used assessment tools. – first you conclude that adapted tests do not have satisfactory psychometric properties, and later you conclude that it is better to adapt a test than to develop it from scratch - unmatched conclusions!
Lines 311-312: additionally, it is important to encourage studies that facilitate the comparison of existing outcomes related to aphasia treatment and rehabilitation – why? This refers more to testing the effectiveness of therapy than to testing validity. It would be more important to compare tests made on the same principles: see for example, Matić Škorić, A., Norvik, M. I., Kuvač Kraljević, J. K., Røste, I., & Simonsen, H. G. (2023). Comprehensive Aphasia Test (CAT): Comparability of the Croatian and Norwegian versions. Aphasiology, 1–27. https://doi.org/10.1080/02687038.2023.2250516 in which authors tested is it possible to create comparable versions of the CAT – the Croatian (CAT-HR) and the Norwegian (CAT-N) - when using the same procedure for test standardisation and validation.
In short, in this paper the authors set themselves up as judges (even though this may not have been their intention) deciding which tests are valid and which are not, without at the same time setting a clear standard for psychometric criteria and explaining why this standard should be always satisfy. If you want to draw such serious conclusions, you should first set the standards, explain them, and then evaluate articles that convey the message to clinicians about what we currently have (not criticizing) and what needs to be changed in order to create more valid and reliable tests in the future.
Authors need to revise the aim and maintain it throughout the paper. Tone down the conclusion. Be more forward-looking by telling authors and journal editors why it is important to publish studies on the validity of diagnostic tools and what these studies should include (i.e. which psychometric measurements). Do not criticize, especially since your review is not comprehensive. Be critical but stimulating.
Author Response
Please see the file attached below.

Reviewer 2 Report
Comments and Suggestions for Authors
Title: The title does not represent what this paper is about. Your work is more about the ‘quality’ of measurement tools, and not the actual measurement properties of the various aphasia tools. Suggest a revision to the title.
Introduction
Lines 38-42 were confusing. Can you please clarify what you are expressing here?
Line 48. Please make a new sentence. Replace ‘instead’ with ‘In contrast’
Line 51: need one word – inaccurate ‘sentences.’
“Outcome measures”: From the start, it would be helpful to clarify the difference between diagnostic tools and outcome measures. It appears that you are summarizing both types of tools. Clarity is needed about what you are trying to capture. We often use diagnostic tools, e.g., Boston Diagnostic Aphasia Test, as an outcome measure. But any of these tools were not developed to be outcome measures. a diagnostic tool to characterize patterns of aphasia. Outcome measures are intended to capture changes associated with intervention for aphasia. Then you need to clarify which of the summarized tools are diagnostic and which are outcome measures.
This introduction does not refer to existing literature from researchers who have tried to establish core outcome sets for aphasia – for example, Wallace et al 2022 Intl J of Lang and Comm Dis.
This introduction would also benefit from an overview of psychometric topics that sound measurement tools would incorporate. In general, the introduction does not set up the purpose of this overall project and needs considerable revamping.
Methods
Line 111: Did you include ‘aphasia’ as part of the search terms for Medline?
2.5 In study selection, please report agreement percentages in the selection of the articles by the two reviewers.
2.6 Provide information about who completed the COSMIN checklist and data about the reliability of scoring between examiners.
Results
Line 187-189 is redundant. You already said that in the methods so no need to repeat.
The results need to more thoroughly summarize the findings of this systematic review. There is a long table with very little summary and interpretation of the findings of the table.
Discussion
Line 248 The end date of this project is not consistent with the end date noted in the methods section.
Line 248: The tables do not include psychometric properties of the varied measurement tools, which is a huge shortcoming of this review. Not sure why the authors are bringing up psychometrics when this was not summarized.
Line 299: This review conflates how used certain tools are with how often studies have examined the psychometric properties of the tool within and across languages. Just because a tool’s psychometric properties were only tested in one study does not correlate with how often it is used in clinical practice.
Writing style: The paper overall would benefit from careful line editing.
Comments on the Quality of English LanguagePlease consider consulting an English editor.
Author Response
Please see the file attached below.

Reviewer 3 Report
Comments and Suggestions for Authors
Dear Authors,
I read your work entitled “Measurement properties of outcome measures for aphasia: a systematic review” and here I enclose my recommendations to you:
1. The “Introduction” section even it is having an amount of information it has a shortage in references. The text is pretty poor, and I suggest the Authors to address those two issues.
2. The same citation issues are also found in the “Discussion” section. It is also noticed that this section is pretty short and not so reader friendly. I suggest the Authors to consider this comment.
Thank you.
Author Response
Please see the file attached below.

Round 2
Reviewer 1 Report
Comments and Suggestions for Authors
I thank the authors for their open-mindedness and willingness to accept all comments. The paper has been greatly improved - the introduction and discussion are better linked, the aim of the paper is clearer, and a nice and constructive message has been conveyed about the importance of following a standardized protocol when reporting the psychometric characteristics of tests. Table 1 and the new title are great.
Once again, I thank the authors and congratulate them on their fine and valuable manuscript.
Author Response
Dear Reviewer,
We sincerely thank you for your positive feedback and for taking the time to review our manuscript. We are pleased that the changes we made improved the clarity and consistency of the work, and that the message about the importance of following a standardized protocol in reporting psychometric characteristics of tests was appreciated.
Your recognition of our efforts to improve the introduction, discussion, and overall organization of the manuscript is very motivating to us. We are especially happy that the new title and the revision of Table 1 met with your favor.
Once again, thank you for your valuable comments and your contribution to the improvement of our work.
Kind regards,
Reviewer 2 Report
Comments and Suggestions for Authors
Thank you for this careful revision. This was a huge reading project that provides very useful information for readers who work with individuals with aphasia. There are just a few remaining edits needed throughout the paper.
Thank for adding the section about ROMA Lines 75-76 were confusing tho. This sentence needs an edit.
Line 196 has a typo in the word ‘cross’
Thank you for adding the section in Methods to define the psychometric concepts.
Table 1: ASHA-FACS needs to include the FACS position: ASHA Functional Assessment of Communication Skills for Adults
Line 243 should say Table 2.
Line 251 and Line 254 and 257: The word ‘used’ here seems not quite right. How about the most widely ‘translated’ or ‘distributed’ tool across languages??
3.4 line 306 needs to be Table 3.
Line 389 should be Table 2.
Line 414: please fix the typo for ‘…were ex did not meet…’
Thank you.
Comments on the Quality of English LanguageMany of my comments speak to remaining writing edits that are needed.
Author Response
Thank for adding the section about ROMA Lines 75-76 were confusing tho. This sentence needs an edit.
Thank you for your comment and for appreciating the addition of the section on ROMA. We have revised and improved the sentence in lines 75-76 to make it clearer and more comprehensible. We hope the new version is more fluent and precise.
Once again, we appreciate your valuable feedback in helping us improve our manuscript.
Line 196 has a typo in the word ‘cross’. the author have modified the word
Table 1: ASHA-FACS needs to include the FACS position: ASHA Functional Assessment of Communication Skills for Adults done
Line 243 should say Table 2.: modified
Line 251 and Line 254 and 257: The word ‘used’ here seems not quite right. How about the most widely ‘translated’ or ‘distributed’ tool across languages??: the term has been modified
3.4 line 306 needs to be Table 3; Line 389 should be Table 2: modified
Line 414: please fix the typo for ‘…were ex did not meet…: done